# FacZ is a GpsB-interacting protein that prevents aberrant division-site placement in *Staphylococcus aureus*

Thomas M. Bartlett[1], Tyler A. Sisley [1], Aaron Mychack[1], Suzanne Walker [1], Richard W. Baker [2,3], David Z. Rudner [1]✉ & Thomas G. Bernhardt [1,4]✉

*Staphylococcus aureus* is a Gram-positive pathogen responsible for antibiotic-resistant infections. To identify vulnerabilities in cell envelope biogenesis that may overcome resistance, we enriched for *S. aureus* transposon mutants with defects in cell surface integrity or cell division by sorting for cells that stain with propidium iodide or have increased light-scattering properties, respectively. Transposon sequencing of the sorted populations identified more than 20 previously uncharacterized factors impacting these processes. Cells inactivated for one of these proteins, factor preventing extra Z-rings (FacZ, SAOUHSC_01855), showed aberrant membrane invaginations and multiple FtsZ cytokinetic rings. These phenotypes were suppressed in mutants lacking the conserved cell-division protein GpsB, which forms an interaction hub bridging envelope biogenesis factors with the cytokinetic ring in *S. aureus*. FacZ was found to interact directly with GpsB in vitro and in vivo. We therefore propose that FacZ is an envelope biogenesis factor that antagonizes GpsB function to prevent aberrant division events in *S. aureus*.

The cell envelope of the opportunistic pathogen *Staphylococcus aureus* is vital for resisting turgor pressure and interfacing with the host. Like most Firmicutes, its surface is composed of a cytoplasmic membrane surrounded by a thick peptidoglycan (PG) cell wall, with anionic polymers called lipoteichoic acids and wall teichoic acids decorating the respective layers[1]. Because the mechanical integrity conferred by the cell envelope is necessary for survival[2], the biosynthetic pathways that build it have been effective targets for many of our most successful antibiotic classes. However, many strains of *S. aureus* have evolved resistance to cell-wall-targeting antibiotics[3]. A deeper understanding of the mechanisms that promote envelope biogenesis is therefore needed to uncover new ways to target this essential structure for drug development.

In rod-shaped bacteria, envelope assembly is performed in two phases: elongation and division. Depending on the organism, elongation proceeds via the insertion of a new cell wall at dispersed sites throughout the cell cylinder or at the cell poles[4]. This phase is followed by cell division, which in Gram-positive bacteria involves the assembly of a multilayered cell wall septum that initially bisects the mother cell. The septum is built by the divisome, a multiprotein PG synthesis machine organized by treadmilling polymers of the tubulin-like FtsZ protein that condense into a dynamic structure called the Z-ring to establish the division site[4]. After the septum is built, enzymes with cell wall cleaving activity, called PG hydrolases or autolysins, split the septum to promote daughter cell separation and complete the division process[5,6].

[1]Department of Microbiology Blavatnik Institute, Harvard Medical School, Boston, MA, USA. [2]Department of Biochemistry and Biophysics, School of Medicine, University of North Carolina at Chapel Hill, Chapel Hill, NC, USA. [3]Lineberger Comprehensive Cancer Center, University of North Carolina at Chapel Hill, Chapel Hill, NC, USA. [4]Howard Hughes Medical Institute, Harvard Medical School, Boston, MA, USA. ✉e-mail: david_rudner@hms.harvard.edu; thomas_bernhardt@hms.harvard.edu

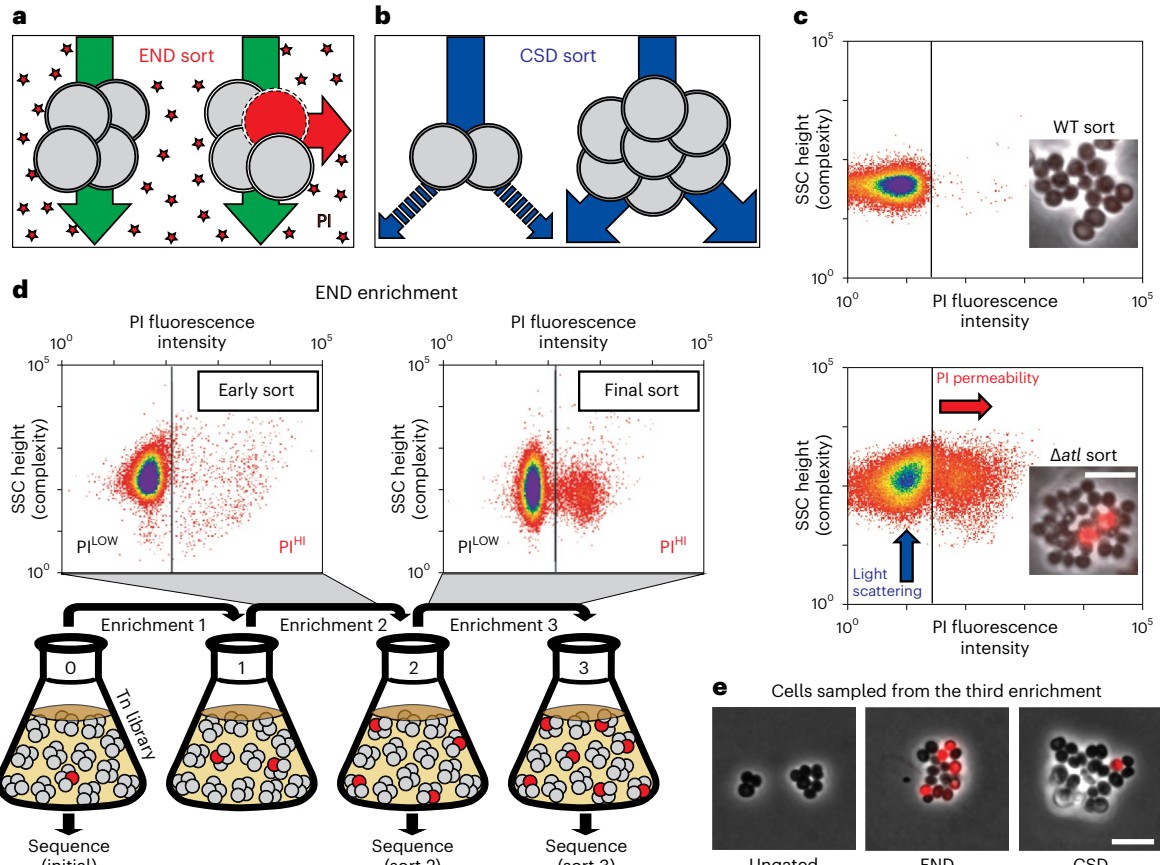

**Fig. 1 | High-throughput FACS-based enrichments for mutants defective in envelope assembly. a,b,** Schematics showing the logic of the END (**a**) and CSD (**b**) enrichments. See text for details. **c,** FACS profiles monitoring light scattering (SSC height) and PI fluorescence for wild type (WT) and Δ*atl* strains. Inset micrographs show phase contrast images with an overlay of PI staining in red (scale bar = 4 µm; the scale bar applies to all images in **c**). **d,** Schematic detailing the END enrichment workflow. An analogous CSD enrichment was performed in parallel (not shown). See text for details. Enrichment for PI^HI mutants was examined by FACS profiling at each stage (top). **e,** The final cell populations (strain aTB015 [RN4220 Δ*attB(f11)::OrfS pTM378*]) from the END and CSD sorts, as well as an ungated control, were imaged on 2% agar pads (scale bar = 4 µm; the scale bar applies to all images in **e**). PI staining (red) was overlaid on the phase contrast image.

Spherical *S. aureus* cells do not have as distinct an elongation phase as their rod-shaped cousins[7]. Instead, envelope biogenesis is principally focused at the division site throughout the cell cycle[8]. In many rod-shaped bacteria, a combination of the Min system and nucleoid occlusion defines the division site by guiding Z-ring formation to midcell[9–11]. Although *S. aureus* also uses a nucleoid occlusion protein[12,13], much less is known about how it regulates division-site placement to localize envelope assembly. As a sphere, *S. aureus* must restrict division to a single midcell plane among the theoretically infinite number of such planes available. An additional complexity in *S. aureus* not shared by rod-shaped cells is that its division site rotates through different perpendicular planes in successive cell cycles[14]. Geometric and/or structural epigenetic cues have been suggested to aid in coordinating division site rotation, but the underlying mechanism is unclear[15,16]. Thus, much remains to be learned about how this seemingly simple, spherical bacterium controls the biogenesis of its envelope.

To uncover new factors involved in envelope assembly in *S. aureus*, we performed two related comprehensive screens for mutants with altered cell morphology and/or increased envelope permeability. Hits included many genes known to be critical for proper envelope assembly and morphogenesis, indicating that the screens worked as intended. The screens also implicated a number of genes of unknown function in envelope assembly, providing a rich dataset for uncovering new insights into the process. We characterized one of these hits, *facZ*, and

found that it encodes a regulator of cell division placement targeting the conserved cell division protein GpsB.

## Results

### Screens for *S. aureus* envelope biogenesis mutants

A high-density transposon mutant library of *S. aureus* (strain RN4220) was constructed and subjected to two enrichment regimens using a cell sorter (Fig. 1). The envelope defective (END) enrichment sorted for mutants with altered membrane permeability based on increased staining with propidium iodide (PI), a membrane-impermeable fluorescent DNA dye (Fig. 1a). Although PI staining is typically associated with lethal loss of membrane integrity[17], we reasoned that we could enrich for mutants experiencing an increased frequency of these defects based on the association of dead, PI-positive cells with viable siblings in unseparated cells. In the related cell separation or division (CSD) enrichment, mutants defective for cell separation or cell division were sorted based on increased light scattering of larger cells or cell clusters[18,19] (Fig. 1b). To set the sorting cut-offs (gates) for each enrichment, we used a Δ*atl* mutant defective in the major cell separation autolysin[5,20] as a control (Fig. 1c). As expected, the larger size and shape complexity of unseparated cell clusters of the mutant gave rise to a population of cells causing increased light scattering (Scatter^HI) compared with a wild-type control. In addition, we observed the appearance of a population of PI-positive (PI^HI) cells in cultures of the Δ*atl* mutant that was largely absent in wild-type cultures (Fig. 1c). The sorting parameters for the

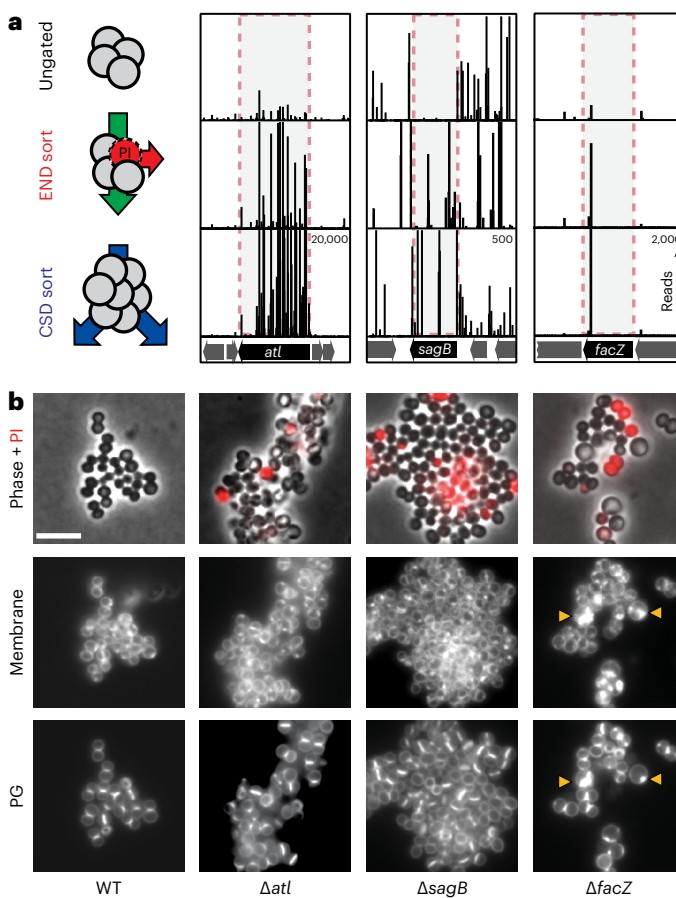

**Fig. 2 | Validation and initial characterization of screening hits. a**, Tn-seq profiles from three genomic loci showing enrichment of transposon insertions at the completion of the END and CSD sorting protocols relative to the control sort. Each vertical line represents a mapped insertion site, and the height of the line is the number of reads mapping to that site, which reflects the representation of the insertion mutant in the population. Profiles for each locus are scaled separately with the maximum number of reads indicated in the top right corner of the bottom profile. **b**, Representative images of WT and mutant cells (Δ*atl* [aTP103], Δ*sagB* [aTB287] and Δ*facZ* [aTB251]) pulse labelled with sBADA to visualize PG synthesis (bottom row), stained with N,N,N-trimethyl-4-(6-phenyl-1,3,5-hexatrien-1-yl)-benzenaminium, 4-methylbenzenesulfonate (TMA–DPH) to label the cell membrane ('Membrane', middle row) and treated with PI to assess envelope permeability ('Phase + PI', top row). Yellow arrowheads highlight membrane and PG synthesis defects. The fluorescence intensity for each channel was scaled identically for all strains to facilitate direct comparison between images (scale bar = 4 μm; the scale bar applies to all images in **b**).

END and CSD enrichments using the transposon library were defined to capture Scatter^HI^ and PI^HI^ cells with similar properties to those of the Δ*atl* mutant, which we anticipated would enrich for mutants with a broad range of envelope biogenesis and cell division defects in addition to those impaired for cell separation (Fig. 1d).

Before sorting, the transposon library was grown to mid-exponential phase. We then collected cells passing the Scatter^HI^ or PI^HI^ gates (10^6 cells for each) as well as an equivalent number of cells sorted without a gate for use as a control. The resulting sorted populations were recovered, amplified in growth medium and sorted again using the same parameters. Three sequential enrichments were performed for each population, maintaining cells in exponential phase throughout the procedure (Fig. 1d). Imaging of cells from the populations recovered following the third sort showed that the protocol was effective in enriching for cells with the desired phenotypes (Fig. 1e).

To monitor changes in the representation of transposon mutants in the library during enrichment, transposon sequencing (Tn-seq) was performed on the original unsorted library and the mutant populations following the second and third sorts for Scatter^HI^, PI^HI^ or the ungated control sort. In the initial transposon library, the median gene had 17 unique insertions, and 2,468 of the 2,629 annotated genes (93.9%) had at least one insertion (Supplementary Table 1). To determine which mutants were being enriched by the sorts, the number of insertion reads at each locus from the END or CSD enrichments was compared with those in the ungated control at the same time point (Supplementary Table 2). By the second sort, we observed a subset of genes showing greater than fourfold enrichment in insertions relative to the ungated control, and this enrichment became even more pronounced by the third sort (Extended Data Fig. 1a,b). Furthermore, for each enrichment, the increased representation of insertions in a given gene following the second sort was correlated to its enrichment following the third sort (Extended Data Fig. 1c,d). Thus, each enrichment was selective, and multiple rounds of sorting yielded stronger phenotypic enrichment.

The enrichments of END and CSD mutants were moderately correlated, with many of the 50 most-enriched mutant loci for each sorting scheme overlapping (Extended Data Fig. 1e and Supplementary Table 3). To identify mutants showing the greatest combined enrichment across both sorts, we analysed the enrichment data using a two-dimensional principal component analysis (2D-PCA; Extended Data Fig. 1f). A total of 74 candidate 'hits' were enriched at least eightfold compared with unsorted controls along the first principal component, PC1, which serves as a proxy for combined enrichment in both screens (Supplementary Table 4). Among the hits were genes with known roles in envelope biogenesis, cell separation and cell division such as *sepF*[21], *atl*[5,20] and *sagB*[22–24], with transposon insertions in these genes showing clear enrichment following both the END and CSD sorts (Fig. 2a, Extended Data Fig. 2a and Supplementary Table 5). Genes of unknown function were also well represented among the list of hits, including *facZ* (*SAOUHSC_01855*), which showed an enrichment in transposon insertions following both sorting methods, and showed aberrant cell morphology and an increased frequency of PI^HI^ cells in cultures (Fig. 2, Extended Data Fig. 2a and Supplementary Table 5). Mutants in genes of unknown function, *SAOUHSC_01975* and *SAOUHSC_02383*, were also identified as hits, and were shown to have defects in cell division and/or envelope permeability (Extended Data Fig. 2). Among these hits, we found *facZ* (*SAOUHSC_01855*) to be particularly interesting because of the unusual nature of the membrane and cell wall depositions observed in cells lacking FacZ (Fig. 2b). We therefore focused on investigating its function.

### *facZ* mutants have pleiotropic envelope defects

To further characterize the envelope defects of cells lacking FacZ, a Δ*facZ* mutant was generated in the HG003 strain background and pulse labelled with the fluorescent D-amino acid (FDAA) (R)-2-Amino-3-(3-(5,5-difluoro-7,9-dimethyl-2-sulfo-5H-4λ4,5λ4-dipyrrolo[1,2-c:2',1'-f][1,3,2]diazaborinin-3-yl)propanamido)propanoic acid (sBADA) to label sites of nascent cell wall synthesis[25], followed by treatment with Nile red and 2-[4-(Aminoiminomethyl)phenyl]-1H-Indole-6-carboximidamide hydrochloride (DAPI) to visualize the membrane and nucleoid, respectively. The mutant cells were more heterogeneous in size than wild-type cells and showed strikingly aberrant membrane and PG accumulations that often colocalized (Fig. 3a–c and Extended Data Fig. 3a). These accumulations excluded the nucleoid, indicating that they projected into the cytoplasmic space (Fig. 3d). Transmission electron microscopy (TEM) also revealed the aberrant accumulation of envelope material in Δ*facZ* mutant cells (Fig. 3a and Extended Data Fig. 3g), and "super-resolution" three-dimensional structured illumination microscopy (3D-SIM) of labelled cells confirmed the colocalization of the membrane and PG stains, and that these projections are probably invaginations that are continuous with the peripheral cell envelope (Extended Data Fig. 3b

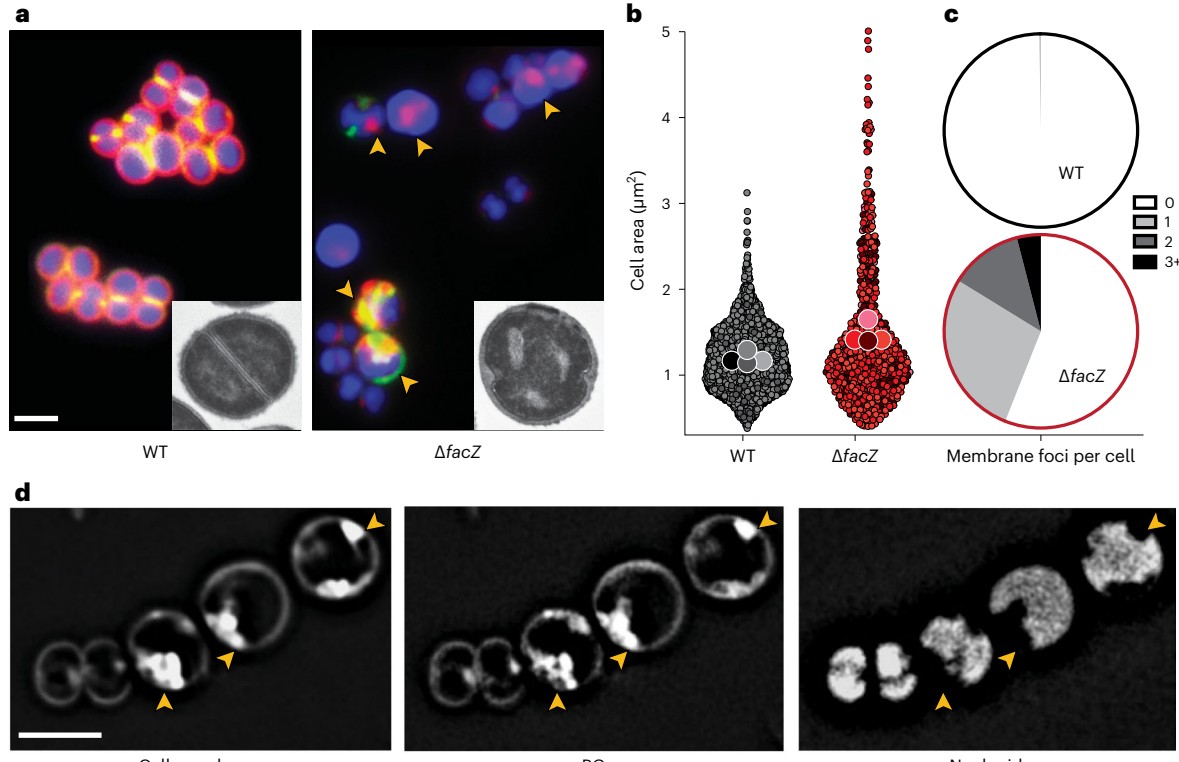

**Fig. 3 | Analysis of morphological defects shown by Δ*facZ* cells. a**, Cells from WT and Δ*facZ* strains were pulse labelled with sBADA to visualize PG synthesis (green), washed three times with PBS to arrest growth and remove unincorporated sBADA, and then labelled for 5 min with the membrane stain Nile red (red). Cells were imaged on 2% agar pads containing DAPI to visualize the nucleoid (blue). Yellow arrowheads highlight cells with aberrant membranes and PG synthesis. Insets show cells from the same strains imaged by TEM. The bright spots in the Δ*facZ* mutant are consistent with the cell envelope inclusions, and probably represent cell envelope projections into the cell interior (Supplementary Fig. 3b). **b**, Violin plots showing the cell area of the indicated strains harbouring a cytoplasmic fluorescent protein. The results from four biological replicates were pooled, each shaded differently; small circles are individual measurements, and large circles are medians from each replicate (WT = 1,760 cells, Δ*facZ* = 701 cells; Supplementary Fig. 3). **c**, Quantification of aberrant membrane foci in TMA–DPH-labelled cells (*n* > 200 cells; Supplementary Fig. 3a). **d**, 3D-SIM microscopy of Δ*facZ* cells stained identically to those in **a**. Yellow arrowheads highlight the position of aberrant cell wall and membrane accumulations that exclude the nucleoid. Scale bars = 2 μm. The scale bars apply to all images in **a** and **d**.

and Supplementary Video). Cells lacking FacZ also showed aberrant localization of FtsZ–GFP. Many cells had multiple FtsZ structures oriented at oblique angles relative to one another, indicative of problems with division site placement (Extended Data Fig. 3c,d). Notably, the essential divisome PG synthesis proteins FtsW and Pbp1 localized to the aberrant envelope invaginations of Δ*facZ* mutants, indicating that the invaginations probably arise from misplaced divisome activity (Extended Data Fig. 3e). Expression of a chromosomally integrated copy of *facZ* under control of the anhydrotetracycline (aTc) inducible *tet* promoter (P*tet*) complemented the morphological and growth defects of Δ*facZ* cells, showing that the defects were caused by the lack of FacZ production and not effects of the deletion allele on the expression of nearby genes (Extended Data Fig. 3f; also see complementation assays in Extended Data Figs. 4g and 5c). We conclude from these data that FacZ is critical for normal envelope biogenesis, and that its absence results in the formation of spurious invaginations of the envelope that represent aberrant, misplaced division attempts.

## Periseptal localization of FacZ

FacZ is an 18 kDa protein predicted to possess an N-terminal transmembrane (TM) helix and a C-terminal cytoplasmic domain composed of a membrane-proximal coiled-coil region followed by an intrinsically disordered region with a net positive charge (Extended Data Fig. 4a,b). AlphaFold Multimer predicts that FacZ forms homo-oligomeric assemblies with three to four protomers with a well-packed interface. In such a configuration, the complex is predicted to be shaped like an anchor, extending ~15 nm into the cytoplasm with disordered C-terminal 'fingers' that extend from a helical rod (Extended Data Fig. 4c).

To determine the subcellular localization of FacZ in *S. aureus*, we constructed a *facZ–mCherry* fusion and expressed it from an ectopic locus under control of the P*tet* promoter in cells that were also pulse labelled with the FDAA sBADA to monitor PG synthesis. Importantly, the fusion protein complemented the growth and morphological defects of Δ*facZ* cells at the same minimal concentration of inducer (50 ng μl⁻¹ aTc) as the analogous untagged allele (Extended Data Fig. 4d–g). All localization experiments were performed using this level of induction for the *facZ–mCherry* fusion. Consistent with its predicted TM helix, FacZ–mCherry showed a peripheral membrane localization, outlining all cells in the population (Fig. 4a and Extended Data Fig. 4d,e). In addition to the peripheral signal, FacZ–mCherry appeared to be enriched at sites of septum formation but confined to the periphery of the septum (the 'periseptum') even at late stages in division (Fig. 4a and Extended Data Fig. 4d,e). We quantified this enrichment by measuring two different fluorescence intensity profiles in populations of labelled cells: one through the cell body perpendicular to the septum (orthogonal profile) and the other around the cell periphery (peripheral profile) (Fig. 4c–f). In cells in the early stages of division identified by their incomplete PG septa, FacZ–mCherry showed enrichment at the edges of the orthogonal profile, and at the periseptum in the peripheral profile, where PG labelling was also most intense (Fig. 4c,d). FacZ periseptal enrichment was slightly but significantly greater (1.18-fold periseptal:peripheral signal, *P* < 0.001)

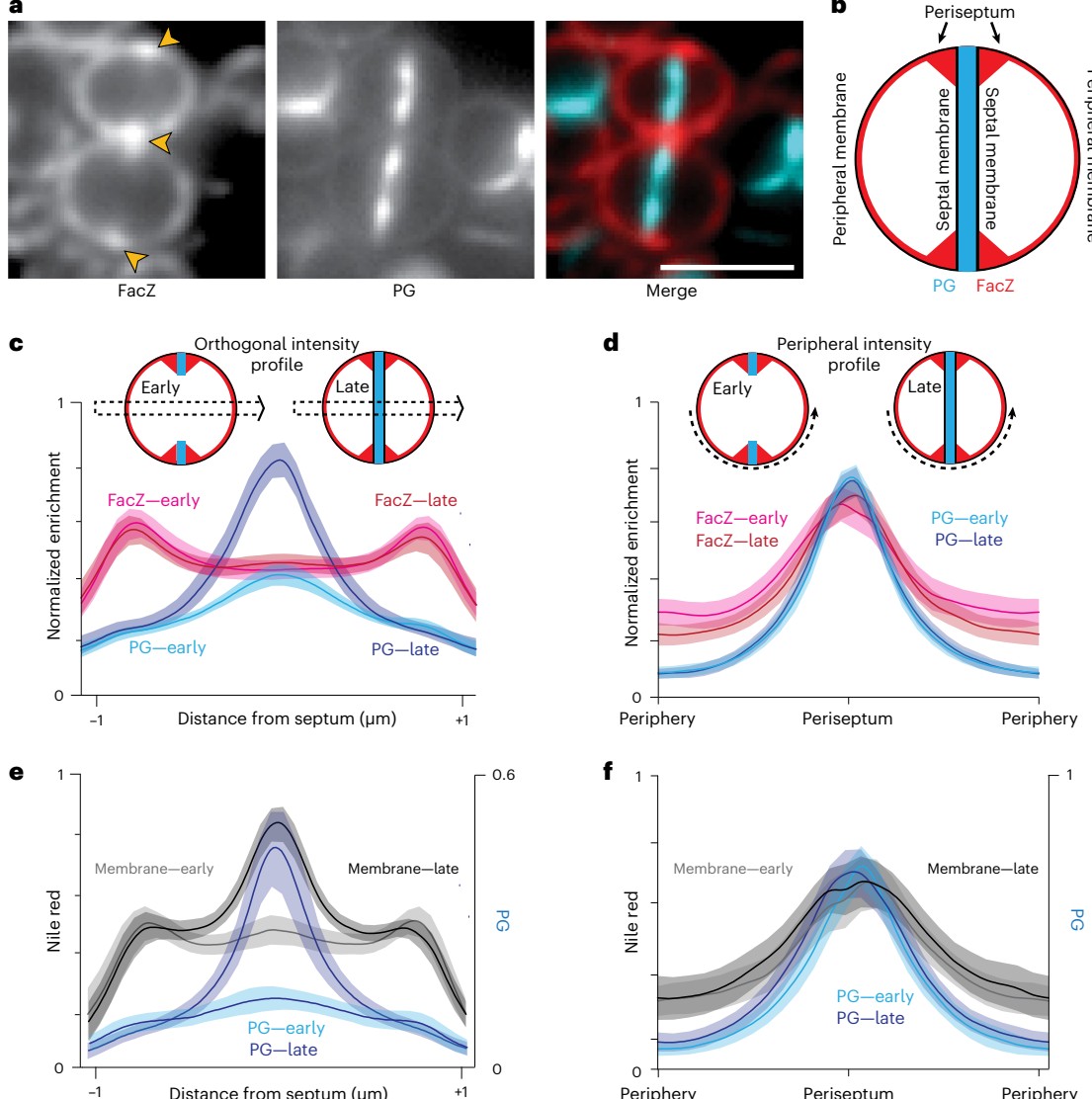

**Fig. 4 | Localization of FacZ in dividing *S. aureus*. a**, Representative fluorescence images of WT cells expressing FacZ–mCherry stained with HADA to label PG (scale bar, 2 μm; scale bar applies to all images in **a**). The protein fusion (left) and PG label (centre) and a pseudo-coloured merged image are shown (FacZ–mCherry in red, HADA in blue) (Extended Data Fig. 5e–g). **b**, Schematic depicting the membrane localization of FacZ in a dividing *S. aureus* cell. **c**, Graphs of mean fluorescence intensity of HADA (light and dark blue) and FacZ–mCherry (light and dark red) collected along lines perpendicular to the septum of cells labelled as in **a**; shaded areas represent 95% confidence intervals. Central peaks show intensity at the septum, and peripheral peaks show intensity at the cell periphery. Cells were grouped into early division (cells with two HADA foci representing an unresolved septum) or late-stage division (cells with a single central HADA focus representing a resolved septum). HADA labelling

is increased in the centre of late-stage-dividing cells (Welch's *t*-test, *P* < 0.05). Localization of FacZ–mCherry does not change significantly as the septum resolves (Kolmogorov–Smirnov test, *P* > 0.05). **d**, Graphs of intensity profile scans measuring fluorescence intensity along peripheral arcs centred on the periseptum for early- and late-stage-dividing cells. **e**,**f**, Graphs of intensity profile scans of WT cells stained with HADA (light and dark blue) and Nile red (light and dark grey) imaged and analysed as in **d** and **e**, respectively. Perpendicular intensity profiles (**c** and **e**) were normalized from 0 to 1 for each cell. Peripheral intensity profiles (**d** and **f**) were interpolated using MATLAB, and intensity profiles were normalized from 0 to 1 for each fluorescence channel within each experiment, to facilitate comparison of cells of different sizes (**c**, *n* = 50 cells; **d**, *n* = 100 cells; **e**, *n* = 24 cells; **f**, *n* = 54 cells).

than a non-specific membrane label (Extended Data Fig. 4f). However, due to the diffraction-limited resolution of our imaging method, the precise localization pattern underlying the periseptal FacZ–mCherry signal remains uncertain (Extended Data Fig. 4e). Regardless, the protein remained strikingly fixed at the periseptum in cells at later stages of division, as the FacZ–mCherry signal in the inner septum remained at or near baseline even in cells in which the septum appeared complete (Fig. 4c). This pattern is in stark contrast to the expected midcell enrichment of the PG and membrane stains in the orthogonal profiles of late-stage septating cells (Fig. 4e). No change was observed in the peripheral profile, as the structure of the periseptum is not expected to

change until the final stage of division when cells separate[26] (Fig. 4d,f). We therefore conclude that FacZ is present and modestly enriched at the periseptum, but unlike typical membrane proteins, it is excluded from the inner septal membrane.

## A role for FacZ in controlling Z-ring formation

To investigate what aspect of cell division FacZ participates in, we used Tn-seq to identify synthetic lethal partners. Transposon libraries were generated in wild-type and Δ*facZ* cells, and the insertion profiles were compared (Supplementary Table 6). Insertion mutations were strongly depleted from many genes in the Δ*facZ* background relative to the wild

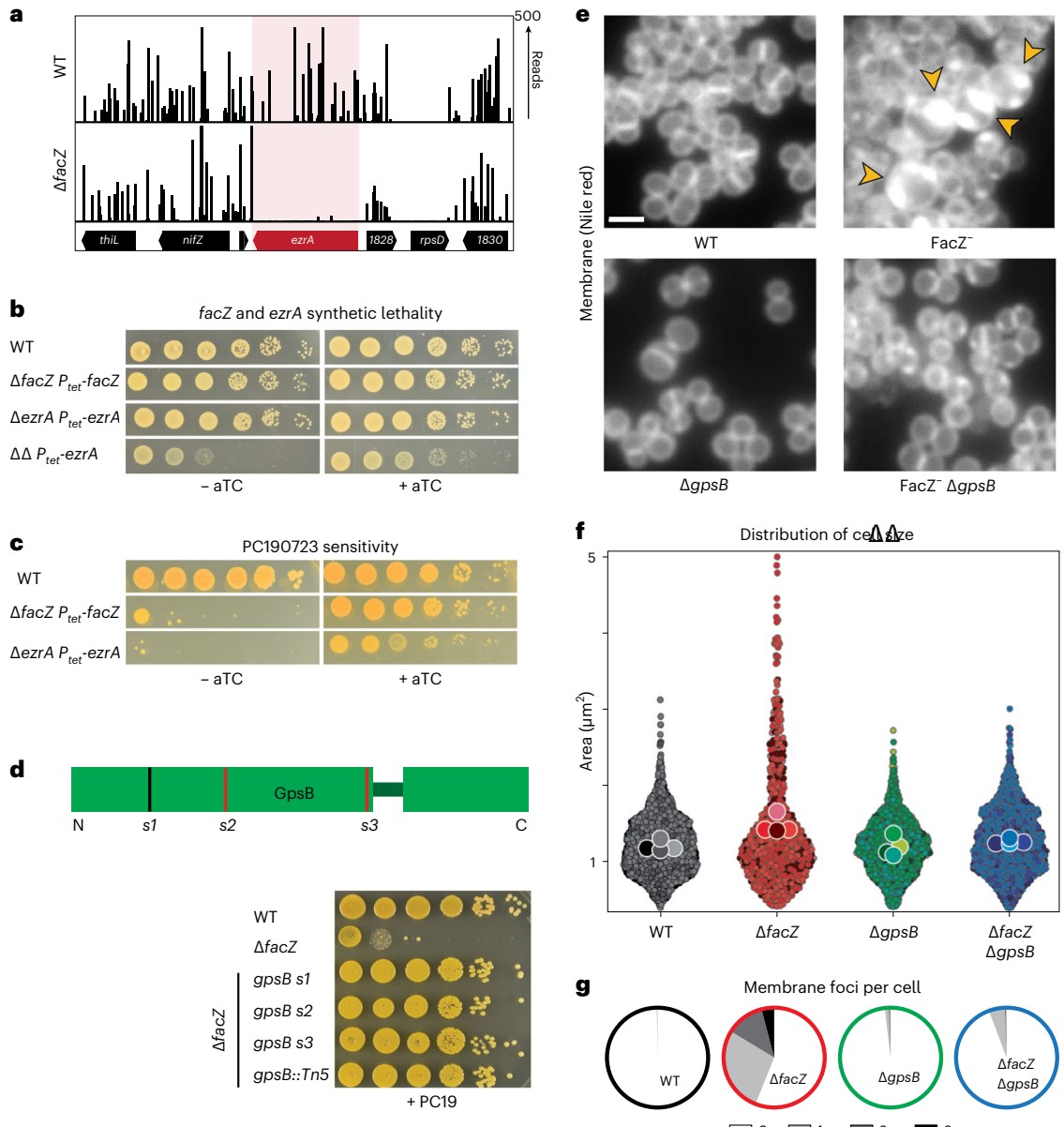

**Fig. 5 | Inactivation of *facZ* impairs cell division and is rescued by deletion of *gpsB*. a**, Transposon insertion profiles of the *ezrA* locus in strains aTB015 [WT] and aTB259 [Δ*facZ*]. **b**, Spot titre of cultures of aTB003 [WT], aTB372 [Δ*facZ* P*tet-facZ*], aTP481 [Δ*ezrA* P*tet-ezrA*] and aTB378 [Δ*facZ* Δ*ezrA* P*tet-facZ*]. Cells were normalized to OD₆₀₀ = 1.0, serially diluted and spotted on TSA agar with or without an aTC inducer (50 ng ml⁻¹). **c**, Spot titre of aTB003 [WT], aTB372 [Δ*facZ* P*tet-facZ*] and aTP481 [Δ*ezrA* P*tet-ezrA*] as in **b** except that the plates contained PC190723 (100 ng ml⁻¹). **d**, Top: Diagram showing the location of mutations in *gpsB* that suppress the PC190723 sensitivity of a Δ*facZ* mutant. Suppressor mutations are mapped onto a diagram of the two folded domains of GpsB (lines indicate position of mutations, with red lines indicating mutations generating premature stop codons). Bottom: Cultures of strains aTB003 [WT], aTB251 [Δ*facZ*], aTB453 [Δ*facZ gpsB*-s1], aTB476 [Δ*facZ gpsB*-s2], aTB478 [Δ*facZ gpsB*-s3] and aTB497 [Δ*facZ gpsB*::*Tn*] were OD normalized and spotted

on TSA supplemented with PC190723 (100 ng ml⁻¹) as in **b**. **e**, Representative fluorescence images of aTB003 [WT], aTB372 [Δ*facZ* P*tet-facZ*], aTB525 [Δ*gpsB*] and aTB540 [Δ*gpsB* Δ*facZ* P*tet-facZ*]. Strains were grown to mid-log phase without induction of *facZ*, and membranes were stained with Nile red (Extended Data Fig. 6). Yellow arrowheads highlight membrane defects. Scale bar = 2 μm. The scale bar applies to all images in **e**. **f,g**, Cultures of aTB521 [WT], aTB527 [Δ*facZ*], aTB529 [Δ*gpsB*] and aTB542 [Δ*facZ* Δ*gpsB*] constitutively expressing cytoplasmic red fluorescent protein from pKK30 were labelled with TMA–DPH in mid-log phase and imaged on 2% agarose pads. **f**, Violin plots showing the cell area of indicated strains based on cytoplasmic fluorescence (WT, *n* = 1,760 cells; Δ*facZ*, *n* = 701 cells; Δ*gpsB*, *n* = 1,544 cells; Δ*facZ* Δ*gpsB*, *n* = 1,217 cells). Cell shape was quantified with MicrobeJ (Methods). **g**, The number of aberrant membrane foci per cell was quantified for display in pie charts (WT, *n* = 791 cells; Δ*facZ*, *n* = 400 cells; Δ*gpsB*, *n* = 998 cells; Δ*facZ* Δ*gpsB*, *n* = 721 cells).

type, with 102 loci bearing a 90% or greater reduction in insertions. Synthetic lethal interactions were identified for genes implicated in several biological pathways. However, many of the genetic interactions were with genes known to be required for cell wall biogenesis and/or cell division (Supplementary Table 7). The *ezrA* gene encoding a regulator of Z-ring formation[27–29] was among these hits (67.6-fold depletion in Δ*facZ*, *P* < 0.005), suggesting that EzrA is essential in the absence of

FacZ (Fig. 5a). We confirmed this synthetic lethal relationship by constructing an EzrA depletion strain and showing that EzrA depletion is lethal in a Δ*facZ* background (Fig. 5b). Notably, mutants lacking either *ezrA* or *facZ* show increased mislocalization of FtsZ–GFP (Extended Data Fig. 3c,d), and both single mutants are hypersensitive to PC190723 (Fig. 5c), a chemical inhibitor of cell division that promotes hyper-stabilization of FtsZ filament bundles in Firmicutes[30].

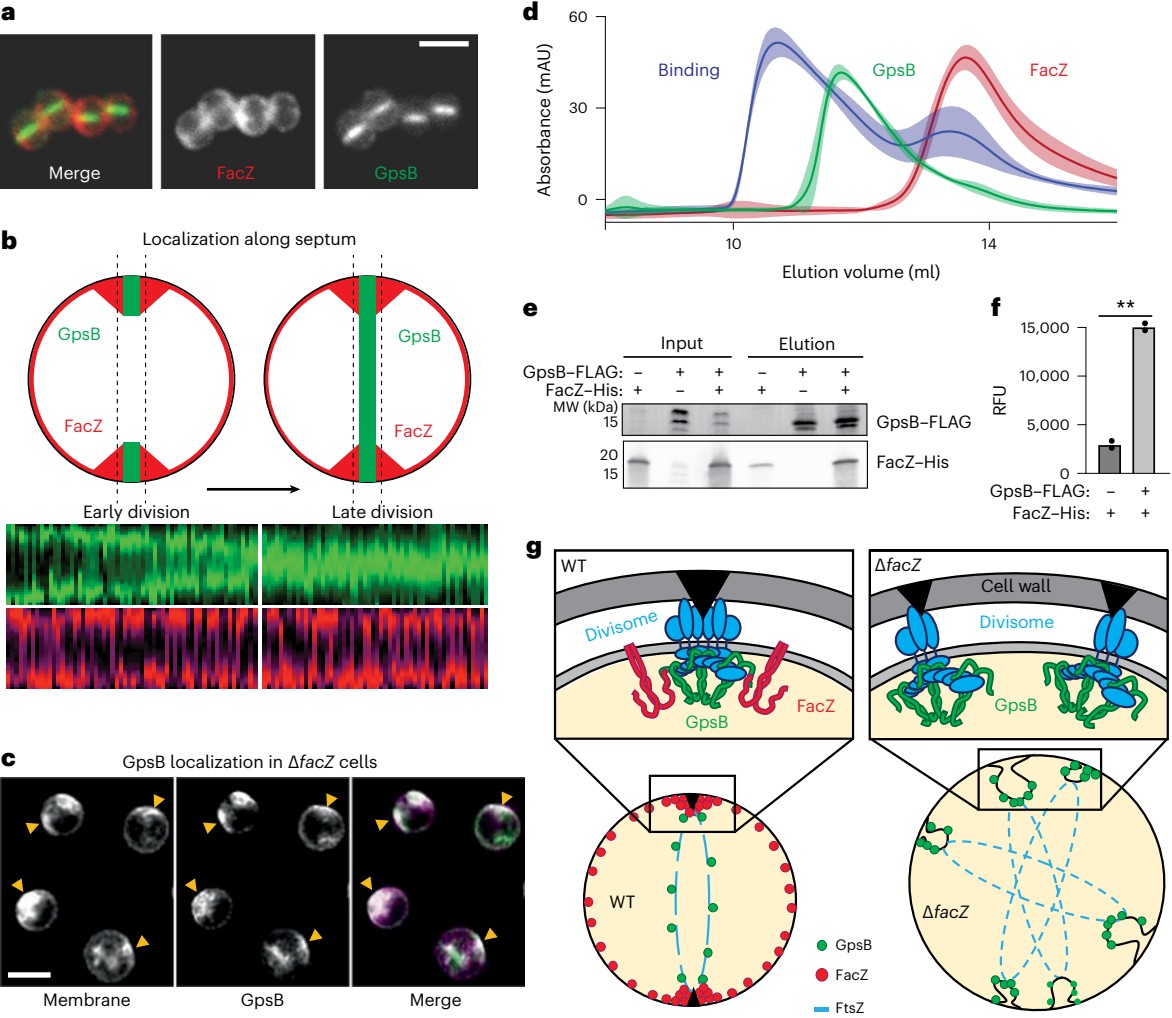

**Fig. 6 | FacZ interacts with GpsB and influences its localization.**
**a**, Representative images of aTB517 [Δ*facZ* Δ*gpsB* P*_tet_*-*gpsB-mNeon* P*_spac_*-*facZ-mCherry*] grown in the presence of aTC (50 ng ml⁻¹) and IPTG (25 ng ml⁻¹) and imaged by fluorescence microscopy (scale bar = 2 µm; the scale bar applies to all images in **a**). **b**, Graphs showing the fluorescence intensity of FacZ–mCherry (light and dark red) and GpsB–mNeon (light and dark green) along lines parallel with the septum. Cells were separated into two groups based on the GpsB–mNeon signal: cells with discontinuous GpsB foci were considered early-division cells (top left) whereas cells with a continuous GpsB–mNeon band at the septum were considered late-division cells (top right) (*n* ≥ 50 cells for each group). **c**, Representative deconvolved fluorescence images of GpsB–mNeon localization in Δ*facZ* cells (aTB519) labelled with Nile red (scale bar = 2 µm; the scale bar applies to all images in **c**). **d**, Gel filtration profiles of GpsB (1–75) alone (green), SUMO-3x-FacZ (127–146) alone (red) or a mixture of the two proteins (binding,

blue). Elution profiles represent the average and standard deviation of six runs. **e**, Lysates were generated from strains expressing tagged variants of FacZ and GpsB [aTB411, 513 and 514]. Membrane fractions were solubilized, loaded onto α-FLAG (DYKDDDDK octamer) magnetic beads, washed and eluted. Input (membrane homogenate) and output (elution) were resolved by sodium dodecyl sulfate–polyacrylamide gel electrophoresis (SDS-PAGE) and blotted using α-His and α-FLAG antibodies (left). **f**, Fluorescence intensities of the FacZ bands from the blot in **e** and a replicate experiment (*n* = 2 biologically independent samples) were measured and relative fluorescence units (RFUs) were plotted (right) (\*\**P* = 0.0018). **g**, Schematic model of FacZ function within the divisome. Cell division is properly localized when FacZ is functional (left). In the absence of FacZ, GpsB is unregulated such that cell constriction is initiated at many sites (right). See text for details.

Based on these genetic results, we infer that FacZ may play a role in controlling Z-ring formation as part of a parallel pathway that is partially redundant with EzrA.

FacZ is conserved in many Firmicutes, including *Bacillus subtilis* (Extended Data Fig. 5a–c). We previously found that inactivation of the *B. subtilis* homologue of FacZ (^Bs^FacZ, formerly called YtxG) resulted in aberrant membrane invaginations and caused defects in sporulation[31]. Our results in *S. aureus* prompted us to reinvestigate the effect of inactivating ^Bs^FacZ in vegetative *B. subtilis* cells. We confirmed that cells deleted for ^Bs^*facZ* show large, aberrant membrane invaginations via Nile red membrane labelling (Extended Data Fig. 5d). In addition, we found that ^Bs^FacZ inactivation caused misplaced FtsZ–GFP structures and led to an increase in cell length, consistent with a division defect (Extended

Data Fig. 5e–h). Thus, FacZ may have a conserved role in promoting proper Z-ring and septum formation among members of the Firmicutes.

**FacZ functions as a GpsB antagonist**
To further explore the role of FacZ in division, we selected for suppressor mutations that overcome the hypersensitivity of *S. aureus* Δ*facZ* cells to PC190723. Whole-genome sequencing of the isolated suppressors identified three unique mutations in *gpsB*, a gene encoding a conserved Firmicute-associated protein with a role in envelope biogenesis[32]. Notably, GpsB in *S. aureus* has been shown to interact directly with FtsZ via a GpsB-interacting motif in FtsZ[33,34] that is not shared by FtsZ from *B. subtilis*[35]. Thus, GpsB is thought to play a direct role in cell division in staphylococci that may be unique to this group of organisms. The *gpsB*

lesions in the Δ*facZ* suppressors included an in-frame duplication of residues 21–24 and two deletions causing premature stop codons at residues 34 and 66, all of which are likely to disrupt GpsB function (Fig. 5d). Although GpsB has been previously described as essential in *S. aureus*[33], Tn-seq analyses from our labs and others have identified Tn mutants throughout *gpsB* in several strains of *S. aureus*[12,36], and the Nebraska Transposon Mutant Library (NTML) collection of non-essential *S. aureus* genes contains a Tn mutant in *gpsB*[37]. Accordingly, we constructed a kanamycin-marked Δ*gpsB* (Δ*gpsB::kan*^R) strain. Both this insertion–deletion and the Tn-inactivated allele from the NTML collection[37] were readily transduced between strains, indicating that *gpsB* is dispensable for growth in our conditions. As expected from the suppressor analysis, inactivation of *gpsB* alleviated the PC190723 hypersensitivity of Δ*facZ* cells (Fig. 5d). Importantly, GpsB inactivation also largely suppressed the cell size and membrane invagination defects caused by deletion of *facZ* (Fig. 5e–g and Extended Data Fig. 6a–d), and partially rescued the synthetic lethal phenotype of Δ*facZ* Δ*ezrA* cells (Extended Data Fig. 6h). These results suggest that FacZ is required to antagonize GpsB function. In support of this idea, overexpression of *gpsB* enhanced the growth defect of the Δ*facZ* mutant (Extended Data Fig. 6i).

Localization of FacZ–mCherry and mNeon–GpsB fusions indicate that both proteins are present at the periseptal region in cells at early stages of division (Fig. 6a,b and Extended Data Fig. 6e–g). However, FacZ–mCherry remains at the periseptum in later stages of division while mNeon–GpsB localizes throughout the septum (Fig. 6a,b and Extended Data Fig. 6e–g). Notably, the localization of mNeon–GpsB was altered upon inactivation of FacZ, with the fusion enriching at many sites throughout the cell coincident with areas of aberrant membrane invagination (Fig. 6c). These findings suggest that FacZ functions by constraining GpsB localization and activity to a single site to promote normal division.

To investigate whether FacZ interacts with GpsB to control its function, we tested whether the two proteins associate in vivo. In *S. aureus* cells, 6xHis-tagged FacZ (FacZ–His) was co-expressed with FLAG-tagged GpsB (GpsB–FLAG). Immunoprecipitation of GpsB–FLAG with anti-FLAG resin resulted in the significant enrichment of FacZ–His in the eluted fraction (Fig. 6e,f and Extended Data Fig. 7f), indicating that the two proteins reside in a complex. To determine whether the interaction is direct, we further characterized their binding in a purified system. Previous crystal structures show that the N-terminal domain of GpsB forms a dimer that binds to small peptides of *S. aureus* FtsZ and PBP4 bearing a (S/T/N)RxxR(R/K) motif[32,35]. We reasoned that FacZ might interact with GpsB by a similar mechanism and found that it contains a canonical staphylococcal GpsB-binding motif at residues 134–139 (NRHYRR) (Extended Data Fig. 5c). Full-length FacZ was insoluble following overexpression. We therefore purified a small ubiquitin-like modifier-tagged (SUMO-tag) fusion bearing a 3× concatenation of residues 127–146 of FacZ (FacZ$_{127–146}$) harbouring the putative GpsB-binding motif. Incubation of this fusion with the N-terminal domain of GpsB (GpsB$_{1–75}$) resulted in a shift in the elution profile of the two proteins by size-exclusion chromatography, indicating that the two proteins interact directly (Fig. 6d). We also found that the arginine residues within the NRHYRR motif are collectively critical for FacZ–GpsB interaction, as the charge-swapped allele NDHYDD is unable to bind GpsB in vitro (Extended Data Fig. 7a,b). Importantly, a FacZ variant harbouring these same changes accumulated to wild-type levels in cells but did not complement the PC190723 sensitivity of a Δ*facZ* mutant in vivo (Extended Data Fig. 7c–e). Thus, the GpsB–FacZ interaction is necessary for FacZ function. These data support a model in which FacZ functions in cell division by directly interacting with GpsB and antagonizing its activity to prevent spurious Z-ring formation and aberrant envelope invaginations (Fig. 6g).

## Discussion

Our genetic screens for envelope biogenesis factors identified FacZ as a new protein important for proper division site placement in *S. aureus*.

FacZ inactivation results in heterogenous morphological phenotypes with a subset of cells showing dramatic division defects. These cells have aberrant, misplaced FtsZ structures and mislocalized invaginations of the cell envelope to which the essential divisome PG synthesis proteins FtsW and FtsI have been recruited. Thus, without FacZ, cells undergo an increased frequency of spurious division events mediated by improperly positioned divisomes. Importantly, the morphological defects observed in a Δ*facZ* mutant are largely suppressed when *gpsB* is deleted. Furthermore, GpsB localizes to the aberrant constrictions formed in the Δ*facZ* mutant, and FacZ directly interacts with GpsB. Thus, our data support a model in which FacZ functions by directly interacting with and constraining the activity of GpsB to prevent aberrant division events.

In *S. aureus*, GpsB was previously shown to interact directly with FtsZ and to promote lateral interactions between FtsZ filaments[33]. GpsB has also been shown to interact with several factors involved in cell envelope biogenesis, including PG synthases and enzymes involved in wall teichoic acid biogenesis[35]. Thus, *S. aureus* GpsB has been proposed to function in the stabilization of Z-rings at the onset of cell division and in the recruitment of enzymes that participate in septum synthesis. It is this cell division activity of GpsB that FacZ is probably antagonizing to prevent aberrant division events. The mechanism by which FacZ restrains GpsB function is currently unknown, but we envision two possible scenarios, both of which rely on the ability of FacZ to directly or indirectly localize to sites where membrane invagination has initiated. The localization and retention of FacZ–mCherry at the periseptum throughout the division process suggests a preference for sites of strong positive Gaussian curvatures such as those formed at the periseptal rim during invagination.

One potential mechanism of FacZ-mediated restraint of GpsB involves the recognition of early signs of septum synthesis by FacZ at positions outside of the preferred division site defined by systems like nucleoid occlusion. In this case, FacZ is expected to interfere with the ability of GpsB to interact with FtsZ and/or other divisome components, potentially through competitive binding, to prevent maturation of an aberrantly placed septum. Alternatively, FacZ could function to reinforce the division site localization signals of other division placement systems by enhancing the ability of GpsB to promote Z-ring formation and divisome maturation at the preferred site. Here the absence of FacZ would cause the aberrant localization of envelope synthesis by reducing the difference in the divisome-forming potential between preferred and non-preferred sites defined by other division site positioning systems. In both cases, we propose that inactivation of GpsB restores normal division by making Z-ring formation and divisome maturation more reliant on signals from other division localization systems that define the preferred site. Further work will be required to differentiate between these and other possible models for FacZ function. Another interesting and outstanding question is why cells have the FacZ–GpsB system in the first place if cell division can proceed relatively normally in the absence of both proteins. Like many other division placement systems in diverse bacteria, these proteins are presumably non-essential owing to functional redundancy, but probably provide a strong selective advantage under some conditions.

The GpsB function has been studied in several organisms in which it has been found to interact with PG synthesis enzymes and influence their localization[34,38], suggesting that it is involved in the spatiotemporal control of their activity. An interaction between GpsB and FtsZ has so far only been detected in *S. aureus*[39]. FtsZ proteins from other Firmicutes such as *B. subtilis* appear to lack a GpsB-interaction motif[35], suggesting that GpsB is not directly controlling Z-ring formation in these bacteria. Similarly, FacZ is unlikely to function in *B. subtilis* by antagonizing GpsB activity as it does in *S. aureus* because ^Bs^FacZ lacks a recognizable GpsB-interaction motif (Extended Data Fig. 5c). Furthermore, we found that inactivation of GpsB did not suppress the sensitivity of a *B. subtilis* Δ*facZ* mutant to PC190723 (Extended Data

Fig. 5i). Thus, although GpsB and FacZ are conserved, and both play roles in envelope biogenesis and division in other organisms, their functional connection to each other and to FtsZ may be restricted to *S. aureus* and its close relatives. Why this functional connection developed specifically in these organisms is not clear, but an attractive possibility is that it arose to provide additional geometrical constraints on division following the rod-to-sphere morphological change during the evolution of staphylococci.

FacZ was just one of many previously uncharacterized proteins that our genetic enrichments have implicated in envelope biogenesis and cell division in *S. aureus*. The dataset we generated should therefore be a valuable resource for uncovering additional new insights into the mechanisms used by this pathogen to control the assembly of its surface. These insights, in turn, will enable future efforts aimed at targeting this essential process for the discovery of novel antibiotics to overcome resistance.

## Methods

### Bacterial growth conditions
All cells were streaked for single colonies on 1.5% agar plates of the appropriate culture medium before the experiments. *S. aureus* strains were grown in tryptic soy broth at 30 °C or 37 °C with aeration, unless otherwise indicated. Cultures were supplemented where necessary with antibiotics at the following concentrations: erythromycin (5 µg ml$^{-1}$ for chromosomal insertions and 10 µg ml$^{-1}$ for plasmids), chloramphenicol (5 µg ml$^{-1}$), spectinomycin (200 µg ml$^{-1}$), kanamycin (kan, 25 µg ml$^{-1}$) and PC190723 (0.2 µg ml$^{-1}$), and induced where necessary with isopropyl β-D-thiogalactopyranoside (IPTG, 25 or 50 µM), or aTc (50 ng ml$^{-1}$ unless otherwise indicated). *B. subtilis* strains were derived from PY79. Cells were grown in Luria broth (LB) or CH medium at 37 °C with aeration, unless otherwise indicated. When necessary, media were supplemented with tetracycline (10 µg ml$^{-1}$), spectinomycin (100 µg ml$^{-1}$), kan (10 µg ml$^{-1}$), chloramphenicol (5 µg ml$^{-1}$), xylose (0.5% w/v) or IPTG (0.5 mM). *Escherichia coli* strains were grown in LB at 37 °C with aeration and supplemented with ampicillin (100 µg ml$^{-1}$). Experiments were conducted on mid-log-phase cells (OD$_{600}$ ~0.4) unless otherwise noted.

### Strain construction
All *S. aureus* deletions were generated in strain RN4220 [WT] by homologous recombination as previously described[12], with the exception of Δ*gpsB::kanR* (TAS201), which was generated via a recombineering system modified from a previous study[40], the details of which will be the subject of a separate publication. Complementation strains were initially made in RN4220 via integration of pTP63 into a phage attachment site on the chromosome of strain aTB033 (RN4220 *pTPO44*)[12], or from the multicopy plasmid pLOW[41] in RN4220. Following initial strain construction, markers were transduced by Φ11 or Φ85 into clean genetic backgrounds. With the exception of the genetic enrichments, all experiments shown were performed in *S. aureus* strain HG003 unless otherwise noted. All *B. subtilis* strains were generated using the one-step competence method unless indicated otherwise. *E. coli* strains used to passage plasmids for *B. subtilis* or *S. aureus* were generated by electroporation or chemical transformation into DH5α cells, while *E. coli* strains used for protein purification were produced by chemical transformation of plasmids into BL21 (DE3). Plasmids were constructed by restriction digest and ligation, or isothermal assembly. Lists of primers, strains, plasmids and descriptions of their construction can be found in Supplementary Tables 8–11.

### Transposon insertion library construction and sequencing
Transposon library construction and sequencing was based on a previous study[12]. Following isolation of cell pellets by centrifugation, cells were lysed with lysostaphin, then treated with RNAse A (ProMega) and Proteinase K (NEB). Genomic DNA was sonicated (Q800R2 QSONICA) to 200–600 bp fragments, then PolyC-tails were added by incubating fragments with a 20:1 mixture of dCTP (Thermo Scientific) to ddCTP (Affymetrix) and TdT enzyme in TdT buffer (ProMega). This DNA was then subject to an initial round of PCR amplification with primers oTB535 and oTB536 with the Easy A cloning kit (Agilent), digested with NotI (NEB) to remove contaminating genomic DNA and subjected to a second PCR amplification with primer oTB537 (an Illumina sequencing primer) and a barcoding primer (unique for each sample to allow for demultiplexing of reads) with the Easy A cloning kit (Agilent). A 200–400 bp product was gel purified and sequenced on an Illumina NextSeq Platform (in the lab) or an Illumina HiSeq 2500 platform (TUCF Genomics Facility, Tufts University). Following sequencing, reads were mapped to reference genome NC_007795 and statistically analysed as previously described[12]. Raw data are available at BioProject ID PRJNA1051644 (http://www.ncbi.nlm.nih.gov/bioproject/1051644).

### Suppressor analysis and whole-genome sequencing
*S. aureus* Δ*facZ* [aTB251] was plated onto restrictive conditions (tryptic soy agar (TSA) supplemented with 0.1 µg ml$^{-1}$ PC190723), and spontaneous suppressors were isolated and cultured overnight at 30 °C, along with the wild type and the parental Δ*facZ* mutant. Cell pellets were collected by centrifugation, then lysed with 20 µg ml$^{-1}$ lysostaphin in phosphate-buffered saline (PBS) at 37 °C for 30 min, and sent to SeqCenter for sequencing (https://www.seqcenter.com/). DNA libraries were prepared using the Illumina DNA Prep kit and Integrated DNA technologies (IDT) 10 bp unique dual indices (UDIs), and sequenced on an Illumina NextSeq 2000, producing 2 × 151 bp reads. Demultiplexing, quality control and adapter trimming were performed with bcl2fastq2 (v2.20.0.422). Variant calling was performed using Breseq (version 0.35.4)[42]. Single nucleotide polymorphisms and deletions were identified by comparing the sequence of the suppressors to that of the parental strains. Raw data are available at BioProject ID PRJNA1051644 (http://www.ncbi.nlm.nih.gov/bioproject/1051644).

### Phenotypic enrichment by Fluorescence-activated Cell Sorting
Fluorescence-activated cell sorting (FACS) was performed with an Astrios FACS (Beckman Coulter) sterilized with bleach and washed with filtered sterile double-distilled water (ddH$_2$O) before and after each sort. For both END and CSD sorts, wild-type *S. aureus* was compared with a Δ*atl* mutant to set phenotypic gates that would exclude normal cells and enrich for mutants with a similar phenotype to the Δ*atl* strain. The Tn library of *S. aureus* RN4220 (described above) was then sorted using these gates alongside a control that was passed through the FACS but without any sorting gates set. The first 10$^6$ cells to pass through the FACS that satisfied the sorting parameters were used to inoculate a 500 ml flask of tryptic soy broth at 30 °C with aeration. Each population was then subjected to iterated rounds of sorting using the same gates between rounds. Cells were kept in exponential phase throughout the sorting, which took place over roughly 36 h. Aliquots from each sort were removed for immediate imaging and frozen in glycerol at −80 °C for Tn-seq analysis.

### Statistical comparison of Tn-seq datasets
Following sorting, sequencing and mapping, the 'read ratio' of each genetic locus was calculated by dividing the number of reads at a given locus in that experimental sort by the number of reads at the same locus in its relevant control population:

$$\text{Read ratio of gene } X$$
$$= \frac{\text{Number of reads in gene } X \text{ in the 'CSD 3' population}}{\text{Number of reads in gene } X \text{ in the 'Ungated 3' population}}$$

Because each dataset contained a different number of reads, these read ratios were not comparable between them. For example, if an experimentally sorted dataset contained more reads than the control

ungated dataset, the mean read ratio for that sort would be greater than one, and this delta would be different for every dataset. To allow comparison between datasets (for example, tracking the change in enrichment of gene X from 'CSD sort 2' to 'CSD sort 3'), the 'relative enrichment' of each gene was determined by subtracting the difference between the 'global' mean read ratio for each population from the 'specific' read ratio at each locus:

Relative enrichment of gene X

= Read ratio of gene X  −  Mean read ratio for relevant sort

For example, if a given gene had a twofold increase in the number of reads in a sorting round, but there were twice as many total reads in that sort than in its ungated control, then its relative enrichment would be 2 − 2 = 0 (zero). These normalized (centred) datasets were used for all comparisons between Tn-seq analyses. A comparison between different rounds of sorting (for example, CSD sort 2 versus CSD sort 3) was performed by linear regression analysis. A comparison between the CSD and END sorts (for example, final CSD enrichment versus final END enrichment) was performed by 2D-PCA. In this analytical reference frame, enrichment along PC1 between rounds of sorting serves as a proxy for joint enrichment in both sorts.

### Fluorescence microscopy

Cells were collected by centrifugation at $3,300 × g$ for 2 min, then immobilized on PBS or M9 agarose pads (2% wt/vol). Cell walls were stained with FDAAs (7-hydroxycoumarincarbonylamino-D-alanine (HADA) or sBADA; Tocris) at 100 μM. Cell membranes were stained with 1 μg ml$^{-1}$ Nile red (ThermoFisher). DNA was stained with DAPI (Molecular Probes) at 2 μg ml$^{-1}$, or with PI (Molecular Probes) at 5 μM. Standard-resolution fluorescence microscopy was performed using a Nikon Eclipse Ti2 or Ti inverted microscope with a Nikon CFI Plan Apo VC ×100 objective lens. Three-dimensional SIM images were acquired with a Nikon Ti2 inverted microscope equipped with an N-SIM Spatial Light Modulator, a Physik Instrument Piezo Z motor and Nikon laser illuminators, and captured with a Dual Hamamatsu Orca Flash 4.0 camera using Nikon Elements 5.11 acquisition software.

### Quantitative image analysis

Image quantification was performed with MicrobeJ[43] in ImageJ. Cell identification, segmentation and morphological assessment were performed on cytoplasmic fluorescence intensity signal. For *B. subtilis*, settings were as previously described[44]. For *S. aureus*, input settings were as follows: background = dark, thresholding = 11, area = 0.4–2.5 and options = clump (0.1); all other settings = default. Parameters were kept the same between different strains and conditions within each experiment. All morphology assessments were fully automated with the following exceptions: categorization of cell cycle phase (early versus late division), number of membrane foci per cell, aberrant versus normal division and angle of division plane, which were manually determined. Furthermore, poorly separated cells that were difficult to segment or obviously misidentified by MicrobeJ were manually removed from analysis. Fluorescence intensity scans (Figs. 4 and 6) were made from user-defined regions of interest. Because fluorescence intensity measurements have arbitrary units, measurements were normalized from 0 to 1 (either within each region of interest, or within all regions of interest measuring the same signal in a given experiment) to allow comparison and graphical display of relative intensity shifts. For measurements of different lengths, measurements were interpolated to facilitate direct comparison between measurements using a custom MATLAB script, which is available on GitHub (https://github.com/DuyuduArtsncrafts/FacZ.git). Fluorescence images were adjusted for display in FIJI and Adobe Illustrator, and all single-channel fluorescence images shown side by side for comparison were collected and analysed with identical parameters.

### Electron microscopy

SEM was performed based on a previous study[12]. Briefly, cells were pelleted and fixed overnight at 4 °C in a mixture of 1.25% formaldehyde, 2.5% glutaraldehyde and 0.03% picric acid in 0.1 M sodium cacodylate buffer, pH 7.4. The fixed tissues were washed with 0.1 M sodium cacodylate buffer and post-fixed with 1% osmium tetroxide and 1.5% potassium ferrocyanide (in $H_2O$) for 2 h. Samples were then washed in a maleate buffer and post-fixed in 1% uranyl acetate in maleate buffer for 1 h. Samples were then rinsed in dd$H_2O$ and dehydrated through a series of ethanol solutions (50%, 70%, 95%, (2×)100%) for 15 min each. Dehydrated tissues were put in propylene oxide for 5 min before they were infiltrated in epon mixed 1:1 with propylene oxide overnight at 4 °C. Samples were polymerized in a 60 °C oven in epon resin for 48 h. They were then sectioned into sections that were 80 nm thin and imaged on a JEOL 1200EX transmission electron microscope. Images were recorded with an AMT 2k CCD camera.

### Protein purification

A DNA sequence encoding a 3× concatenation of FacZ$_{127-145}$ separated by GSAG linkers was gene synthesized and cloned into a vector with an N-terminal His−SUMO tag, with the sequence EIADKWQNRHYR-RGSANYKAGSAGEIADKWQNRHYRRGSANYKAGSAGEIADKWQNRHYR-RGSANYKA (linker sequence underlined). Amino acids 1–75 of *S. aureus* GpsB were cloned into a vector bearing an N-terminal His−SUMO tag. His−SUMO−FacZ$_{127-145}$ and His−SUMO−GpsB$_{1-75}$ were purified separately using the same protocol. A 1:100 dilution of an overnight culture of BL21 (DE3) cells carrying expression plasmid was inoculated into LB kan (50 μg ml$^{-1}$; 1 l), grown to OD$_{600}$ = 0.6 at 37 °C and induced with 0.5 mM IPTG at 18 °C for 18 h. All following steps were performed on ice or at 4 °C. Cells were collected by centrifugation and lysed via sonication in 40 mM Tris pH 8, 500 mM NaCl, 10% glycerol and 10 mM imidazole, supplemented with 1 mM dithiothreitol (DTT), 3 mM phenylmethylsulfonyl fluoride and 1 mM benzamidine. Lysates were clarified via centrifugation and applied to 4 ml of pre-equilibrated Ni$^+$-NTA resin (GoldBio) in a gravity flow column, washed extensively and eluted in 300 mM imidazole. For GpsB$_{1-75}$, the SUMO tag was cleaved with SUMO protease, followed by reapplication to Ni$^+$-NTA resin to remove uncleaved protein and the His−SUMO tag. Proteins were dialyzed against a buffer containing 20 mM Tris pH 8, 500 mM NaCl, 10% glycerol and 1 mM DTT for >24 h, concentrated to >10 mg ml$^{-1}$ and frozen at −80 °C.

### GpsB and FacZ gel-filtration analysis

His−SUMO−FacZ$_{127-145}$ and GpsB$_{1-75}$ were thawed from frozen aliquots and separately buffer exchanged into an identical buffer (20 mM 4-(2-hydroxyethyl)-1-piperazineethanesulfonic acid (HEPES), pH 7.6, 500 mM NaCl, 1 mM DTT) using PD10 desalting columns (GE Healthcare). Gel filtration samples were made by mixing GpsB$_{1-75}$ at 90 μM with SUMO−FacZ$_{127-145}$ at 30 μM in 500 μl reactions. Concentrated protein stocks were diluted with HEPES buffer lacking salt to reduce the salt concentration to a final concentration between 150 mM and 200 mM. Samples were fractionated on a Superdex75 Increase 10/300 gel filtration column (Cytivia) equilibrated in a buffer containing 20 mM HEPES, pH 7.6, 100 mM NaCl and 1 mM DTT (flow rate = 0.5 ml min$^{-1}$).

### FacZ conservation analysis

To show conservation of FacZ homologues, DUF948 was mapped to representative dendrograms of bacteria using AnnoTree version 1 (ref. 45). For these analyses, the Firmicutes (also called Bacillota) were defined as the largest phylum-level monophyletic group containing the families Bacillaceae, Staphylococcaceae and Streptococcaceae. Multiple sequence alignments were carried out using Clustal Omega version 1.2.4 (ref. 46).

### FacZ structural predictions

A simple assessment of FacZ topology was performed with Protter[47]. Characterization of the coiled-coil region was performed with the

COILS server[48], and identification of the C-terminal intrinsically disordered region was performed with IUPRED2A[49]. Homo-oligomeric assembly predictions were made using AlphaFold Multimer v2.1.2 (ref. 50), based on standard output metrics (overall predicted Local Distance Difference Test (pLDDT) score, predicted Template Modeling (pTM) score, interface predicted Template Modeling (ipTM) score).

## Spot titre
Overnight cultures of *S. aureus* were normalized by optical density (OD$_{600}$ = 1.0) and subject to tenfold serial dilution. A 3 µl volume of each dilution was spotted onto TSA agar supplemented with antibiotics or inducers when necessary. Plates were incubated at 30 °C or 37 °C overnight and imaged the next day.

## Statistics and reproducibility
Unless otherwise noted, experiments were carried out in at least biological triplicate, and images shown are representative of multiple experiments. Graphs either show the results of representative experiments, with error bars addressing uncertainty within an experiment, or they show the median and mean of multiple experiments, with error bars addressing the uncertainty between experiments, as indicated in the figure legends. Specifically, all wide-field microscopy analyses were carried out in at least biological triplicate, and many cells from multiple fields of view were analysed. All spot titres were carried out in at least biological triplicate. In vitro binding assays were performed in triplicate using proteins from two independent purifications (six total replicates) for all three conditions assayed (FacZ alone, GpsB alone and FacZ + GpsB). Electron microscopy and 3D-SIM imaging were not reproduced, as they primarily serve to improve image resolution for display purposes and, in all cases, confirm the findings of successfully replicated wide-field experiments. Similarly, genetic screens were not directly replicated, but all hits were confirmed by genetic analysis, and phenotypes were confirmed and replicated at least three times. In general, *P* values were derived from Student's *t*-tests when comparing reasonably normal distributions of nearly equal variance, Welch's *t*-tests when variance was not similar, and Kolmogorov–Smirnov tests when distributions were non-normal. Unless otherwise noted, all *t*-tests were two tailed.

## Reporting summary
Further information on research design is available in the Nature Portfolio Reporting Summary linked to this article.

## Data availability
Code for interpolating lengths of regions of interest for image analysis is available on GitHub (https://github.com/DuyuduArtsncrafts/FacZ.git). Raw sequencing data are available for download at BioProject ID PRJNA1051644 (http://www.ncbi.nlm.nih.gov/bioproject/1051644). An uncropped version of the Western blot in Extended Data Fig. 7e with a molecular weight ladder is available for download (https://doi.org/10.6084/m9.figshare.24803361.v1). All raw data for graphs are supplied in source data files. Source data are provided with this paper.

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

## Acknowledgements

We thank the Bernhardt and Rudner labs for support and discussion, especially A. Vettiger, J. D. Amon, J. Flores-Kim, E. M. Fivenson and K. E. Hummels; the Walker lab for *S. aureus* expertise, especially J. E. Page and M. C. Stone; C. Henriksen for helpful discussion; J. W. Sher for data analysis resources; and S. V. Owen for advice and support. Fluorescence microscopy was performed at the Microscopy Resources on the North Quad (MicRoN) core at Harvard Medical School; electron microscopy imaging was performed with HMS Electron Microscopy Facility. T.M.B. is a jointly mentored postdoctoral fellow bridging work in both the Bernhardt and Rudner labs and was funded in part by the Ruth L. Kirschstein Postdoctoral Individual National Research Service Award from the National Institutes of Health (NIH-NIAID, F32AI150002). This work was also supported by the Howard Hughes Medical Institute (T.G.B.) and the National Institutes of Health grants R01AI083365 (T.G.B.), R01GM127399 and R01GM086466 (D.Z.R.), R01AI139083 (T.G.B. and D.Z.R.) and U19 AI158028 (T.G.B., D.Z.R. and S.W.).

## Author contributions

Project conception and management were performed by T.M.B., D.Z.R. and T.G.B. R.W.B. designed, performed and analysed the structural predictions of FacZ and the in vitro study of the FacZ–GpsB interaction in consultation with T.M.B. T.A.S. designed, performed and analysed the in vivo GpsB–FacZ co-immunoprecipitation and protein blotting in consultation with T.M.B. All other experiments were designed, performed and analysed by T.M.B. with guidance from D.Z.R. and T.G.B. A.M., T.A.S. and S.W. contributed unpublished strains and genetic resources. The paper was written by T.M.B., R.W.B., D.Z.R. and T.G.B.

## Competing interests

The authors declare no competing interests.

## Additional information

**Extended data** is available for this paper at https://doi.org/10.1038/s41564-024-01607-y.

**Correspondence and requests for materials** should be addressed to David Z. Rudner or Thomas G. Bernhardt.

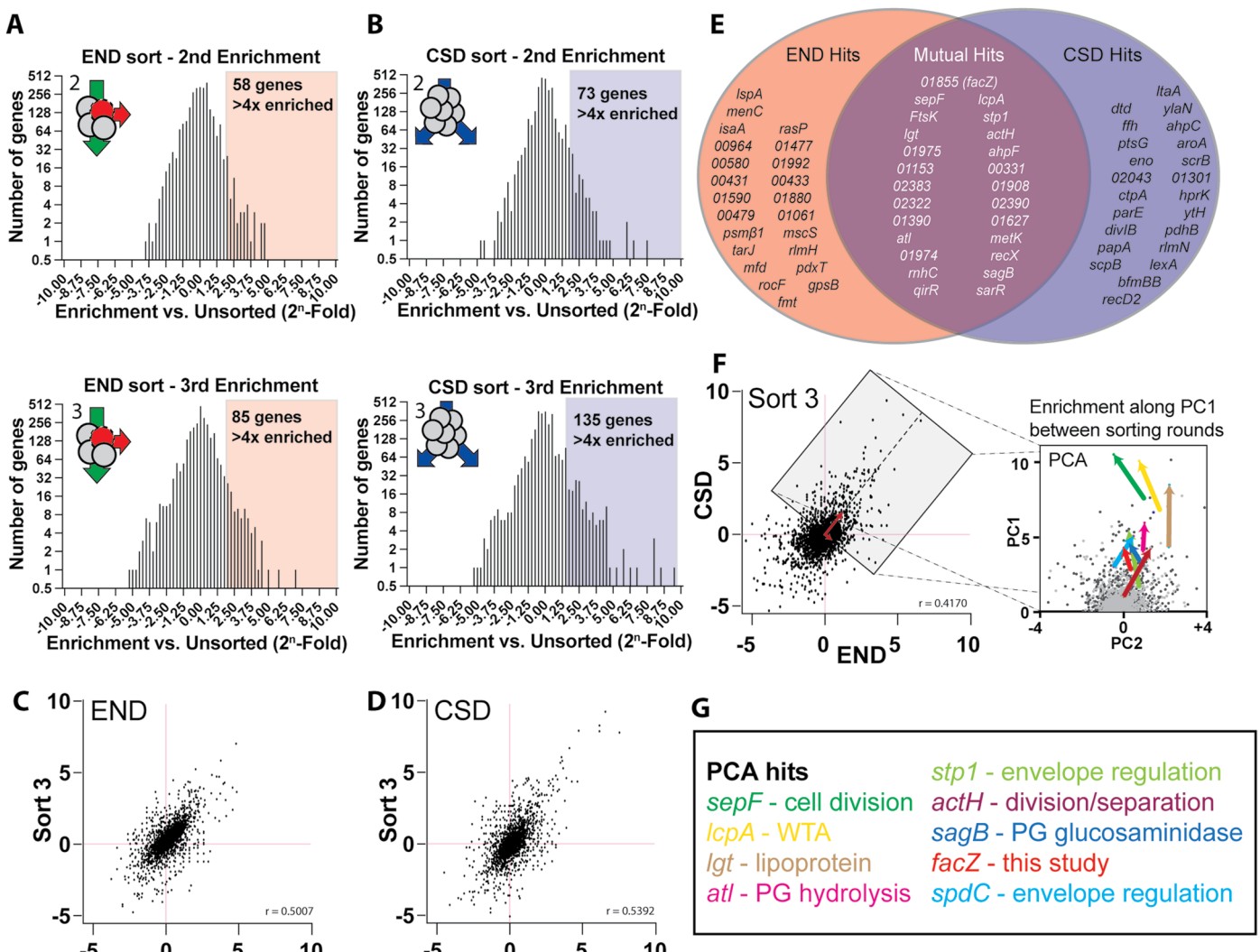

**Extended Data Fig. 1 | CSD and END enrichment data.** (**a**, **b**) Histograms showing the relative enrichment of transposon insertions in genes following the second and third END (A) or CSD (B) sorting rounds. Colored boxes highlight genes with >4x enrichment in transposon insertions. (**c**, **d**) Scatterplots showing relative enrichment at each locus in sort 3 over relative enrichment at the same locus in sort 2 for both the END (C) and CSD (D) screens. Linear regression analysis indicates that relative enrichment is well-correlated between sorting rounds, suggesting enrichment is driven by phenotypic selection and not chance. (**e**) Venn diagram comparison of the 50 genes in which transposon insertions were most strongly-enriched in the CSD and END sorts. (**f**) Enrichment in the final CSD sort is moderately correlated with enrichment in the final END sort, consistent with the overlapping mutant populations isolated from the two screens. 2D-PCA of the final sorts identified a vector (PC1, red) that serves as a proxy for enrichment in both sorts, allowing ranking of mutual hits based on enrichment along this vector between rounds of sorting (PCA panel, and see Supplementary Tables 4–5). The PCA pop-out panel (right) shows relative enrichment at each genetic locus in round 2 (light gray) and round 3 (dark gray), rotated so that PC1 is vertical. Colored arrows show the change in enrichment between rounds of sorting for a subset of hits with known roles in envelope biogenesis. Upward movement in this vector space indicates enrichment in both screens. (**g**) A selection of hits identified in the PCA, many of which are known cell-envelope biogenesis genes. Text colors match with the arrow colors for the given genes in the PCA plot.

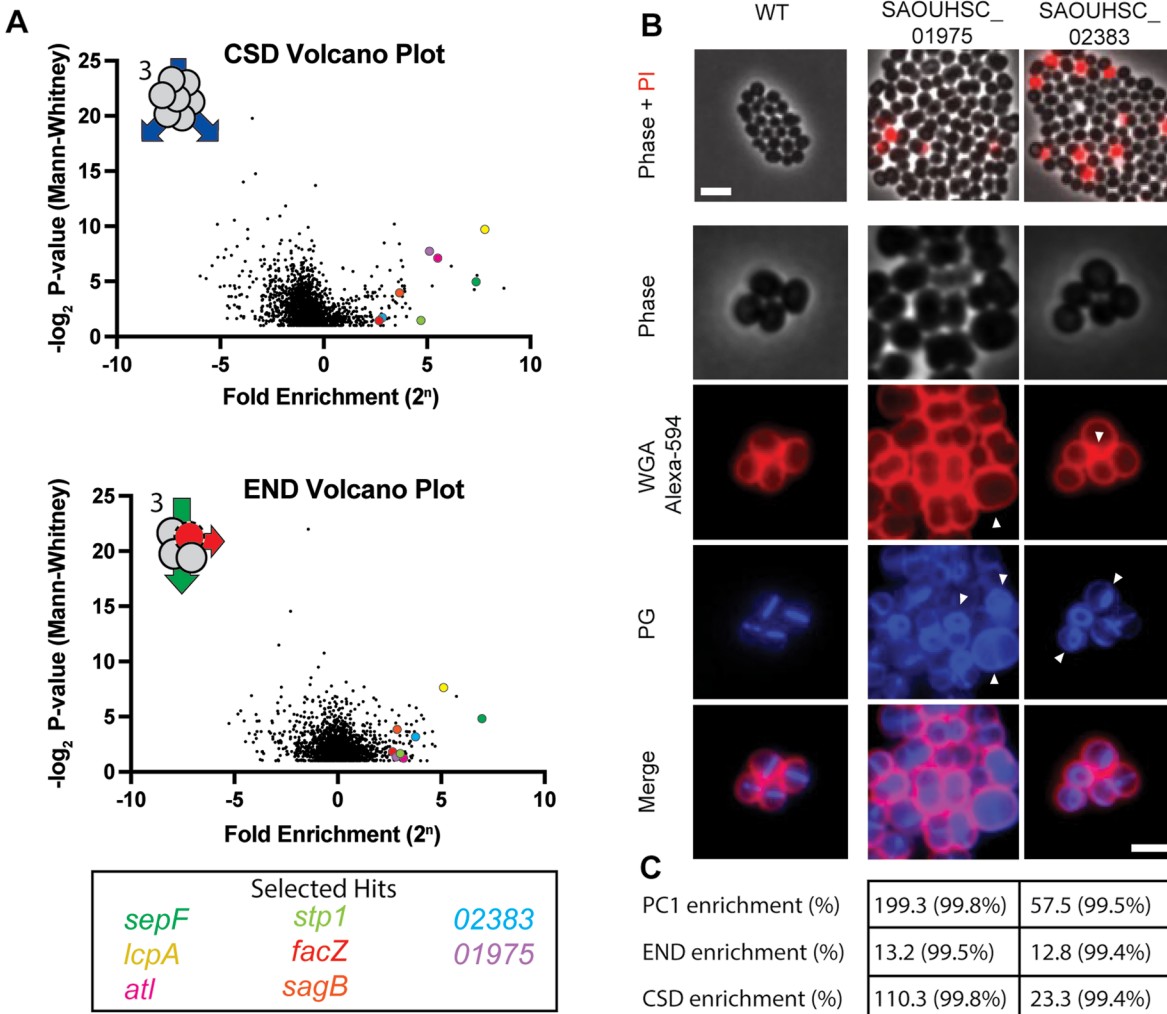

**Extended Data Fig. 2 | Validation and characterization of additional hits from the screen. a**) Volcano plots showing the relative enrichment of transposon insertions in each locus versus the p-value from Mann-Whitney U for the same locus. The final CSD sort (top) and final END sort (middle) are compared to an ungated control generated in parallel. Selected hits (bottom) are color coded and correspond to the colored circle in the plots. **b**) Representative images of two mutants identified in our screens, derived from the NTML Tn-mutant library. To assess END phenotype (top), mid-log phase cells were washed with PBS and incubated for 5 minutes with PI, then placed on pads containing PI and imaged. To assess CSD phenotype (bottom), mid-log phase cells were pulse-labeled with HADA to visualize peptidoglycan synthesis (PG) (middle row, blue), washed three times with PBS to arrest growth and remove incorporated HADA, and then labeled with WGA Alexa-594 to label teichoic acid (top row, red). Compared to the parental *S. aureus* USA300 strain [aTB001], Tn-inactivation of USA300 orthologs of SAOUHSC_01974 [aTB112] and SAOUHSC_02383 [aTB113], have increased PI staining and morphological defects (white arrowheads), consistent with END and CSD phenotypes. Scale bars: top = 2 μm, bottom = 1 μm. **c**) Table showing the fold-change of transposon insertions in the genes inactivated in (B) in the final round of the END and CSD sorts, as well as relative enrichment along PC1. The enrichment percentile of mutations in those genes compared to all mutants is given in parentheses.

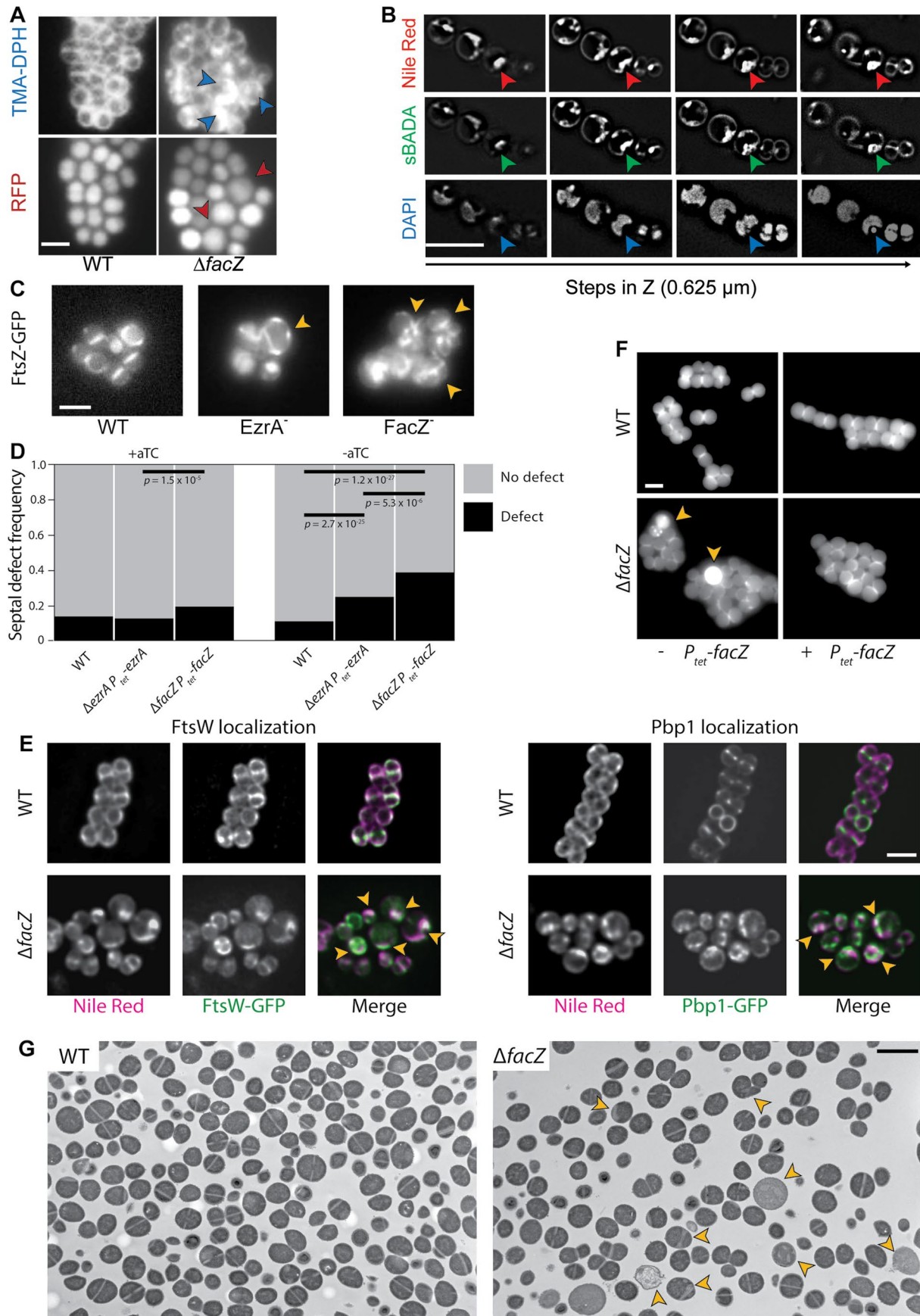

**Extended Data Fig. 3 | See next page for caption.**

**Extended Data Fig. 3 | Inactivation of FacZ impairs cell division and morphogenesis. a**) Representative images showing membrane labeling and cytoplasmic fluorescence of WT [aTB523] and Δ*facZ* [aTB527] cells used to quantify morphological and membrane defects in Fig. 3b, c. Cells were grown overnight in the presence of Tmp (5 µg/ml) to maintain the RFP-bearing plasmid, then subcultured into medium free of antibiotics, and grown into mid-log phase. These cells were then labeled with TMA-DPH, imaged on M9 pads (2% agarose), and segmented on cytoplasmic fluorescence signal. Cell size measurements for violin plots (Fig. 3b) were automated using MicrobeJ; red arrowheads point to unusually large cells. For each cell identified by MicrobeJ, the number of aberrant membrane foci (blue arrowheads) was recorded manually (Fig. 3c). **b**) Z−stack of 3D-SIM reconstruction of Δ*facZ* cells [aTB251] stained identically to Fig. 3 shows aberrant features. Red arrowheads follow membrane features (Nile Red, top) through Z−planes (left to right); features are continuous throughout the cell in the Z−dimension, and do not emerge in the middle of the cytoplasm, consistent with these membrane features being continuous invaginations of the cell membrane, rather than completely internalized structures. Analogous continuous features are apparent when imaging labeled cell wall (green arrowheads, sBADA, middle), and correspond to local exclusion of the nucleoid (blue arrowheads, DAPI, bottom), both of which also extend throughout all Z−slices. **c**) FtsZ-GFP was induced at low levels in exponentially-growing *S. aureus* to determine which cells were dividing and to determine the orientation of the division plane. In WT *S. aureus* [aTB219], most cells exhibited a single-ring, consistent with normal cell division. In cells depleted of FacZ [aTB390], aberrant FtsZ structures (defined as multiple Z-structures, drastically off-centre Z-structures, or diffuse cytoplasmic

FtsZ-GFP signal) were apparent. A similar phenotype was observed in cells depleted for the division protein EzrA [aTB391]. **d**) Stacked bar graphs showing the frequency of aberrant FtsZ structures (black) and normal FtsZ structures (gray) from a representative experiment, with horizontal bars indicating significant differences (p-value < 0.001). Left: The division defect associated with inactivation of FacZ was largely rescued in the presence of aTC (25 ng/mL) to induce *facZ* expression (Δ*facZ* $P_{tet}$-*facZ* vs. Δ*ezrA* $P_{tet}$-*ezrA*: $p = 1.5 \times 10^{-5}$, chi-squared test); WT: n = 129 cells, Δ*ezrA* $P_{tet}$-*ezrA*: n = 100 cells; Δ*facZ* $P_{tet}$-*facZ*: n = 114 cells. Right: Depletion of EzrA or FacZ causes division defects (WT vs. Δ*ezrA* $P_{tet}$-*ezrA*: $p = 2.7 \times 10^{-25}$; WT vs. Δ*facZ* $P_{tet}$-*facZ*: $p = 1.2 \times 10^{-27}$; Δ*facZ* $P_{tet}$-*facZ* vs. Δ*ezrA* $P_{tet}$-*ezrA*: $p = 5.3 \times 10^{-6}$, chi-squared test); WT: n = 105 cells, Δ*ezrA* $P_{tet}$-*ezrA*: n = 143 cells; Δ*facZ* $P_{tet}$-*facZ*: n = 100 cells. **e**) Micrographs showing localization of divisome PG synthases FtsW-GFP and GFP-Pbp1 expressed from a multicopy pLOW plasmid in WT and Δ*facZ* cells as described (see Methods). Left: FtsW-GFP localizes to the divisome in WT cells [aTB666] but mislocalizes to the envelope foci characteristic of Δ*facZ* mutants [aTB675]. Right: similar localization patterns are observed with GFP-Pbp1 in WT [aTB665] and Δ*facZ* cells [aTB673]. **f**) WT *S. aureus* with [aTB341] and without [aTB003] an integrated $P_{tet}$-*facZ* expression construct were imaged alongside Δ*facZ* cells with [aTB372] and without [aTB251] the same integrated *facZ* expression construct. Cells were grown to mid-log phase, treated with aTC (25 ng/mL) to induce FacZ expression from the $P_{tet}$ promoter, and exposed to HADA to label active zones of PG insertion. Only Δ*facZ* cells lacking the complementing allele exhibited morphological defects (yellow arrowheads). **g**) Representative full fields of view corresponding to electron micrograph conditions displayed in Fig. 3a insets. (All scale bars = 2 µm).

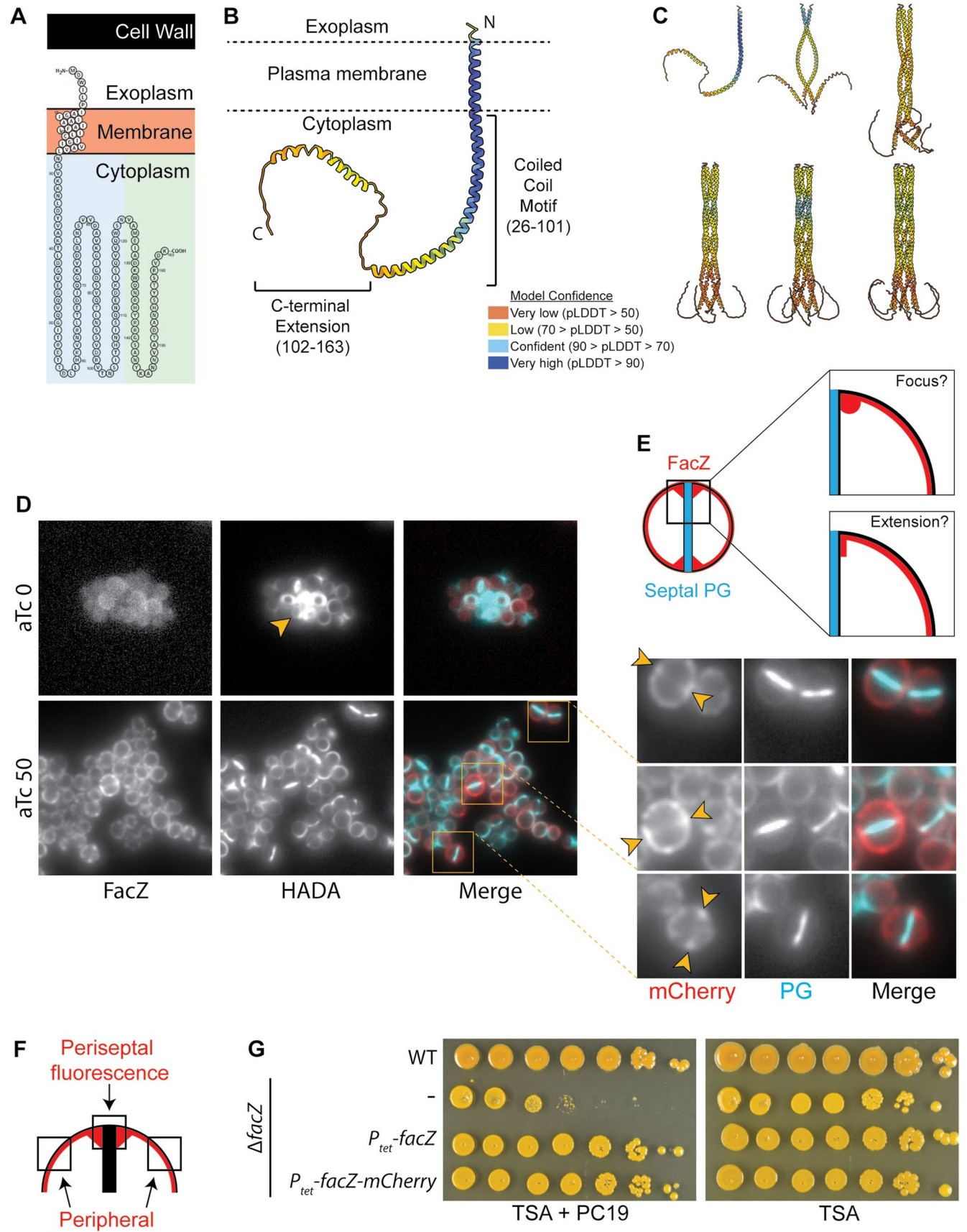

**Extended Data Fig. 4 | See next page for caption.**

**Extended Data Fig. 4 | FacZ structural predictions and functionality of the FacZ-mCherry fusion. a**) Predicted membrane topology of FacZ (Protter)[1]. FacZ is predicted to have one transmembrane helix with a short extracellular N-terminal region. Most of the protein is predicted to extend into the cytoplasm, with a coiled-coil region (blue box) and a disordered C-terminal extension (green box). **b**) AlphaFold2 model of full-length FacZ, colored by pLDDT confidence score. Predicted domains and orientation in the plasma membrane are labeled. **c**) AlphaFold2-multimer predictions of FacZ assemblies from one copy to six copies. **d**) A Δ*facZ* strain harboring *facZ-mCherry* under control of a $P_{tet}$ promoter [aTB373] was grown into mid-log phase with 0 or 50 ng/μl aTC, pulse-labeled with HADA as described, and imaged on PBS pads containing 2% agarose. Cells depleted of or overexpressing FacZ-mCherry (red) had aberrant sites of PG synthesis (yellow arrowheads), while aTC (50 ng/mL) induction of *facZ-mCherry* restored normal morphology and PG incorporation. Insets, bottom right: When FacZ-mCherry is expressed at levels that restore normal cell morphology to Δ*facZ* cells, the fluorescent fusion is enriched at periseptal regions, flanking actively-growing zones of septal PG synthesis (yellow arrowheads) (scale bar = 2 μm). **e**) Diagram showing two potential models for FacZ-mCherry localization at the periseptum. FacZ may be enriched at the highly-curved periseptal rim, or may be equally enriched along the curved peripheral surface and extend slightly

into the flat septal membrane. **f**) Schematic showing regions of cell surface used to measure periseptal:peripheral fluorescence ratios. Periseptal regions were defined as the six pixels centred around the septum (the peak of HADA labeling in dividing cells). The peripheral regions were defined as the six pixels half the distance between the periseptum and the ends of hemispherical arcs along which fluorescence intensity was measured. These intermediate regions were chosen because the extreme ends of the hemispherical arcs could display increased membrane abundance due to previously-formed septa in cells with immature division planes, or nascent orthogonal septa in cells with completed division planes. Periseptal enrichment was measured by taking the ratio of periseptal fluorescence intensity to peripheral fluorescence intensity for FacZ-mCherry (2.10-fold periseptal enrichment) or Nile Red (1.79-fold periseptal enrichment), a non-specific membrane label (see Fig. 4). FacZ-mCherry periseptal enrichment was 1.18 times greater than Nile Red periseptal enrichment (Student's *t*-test, $p = 2.8 \times 10^{-4}$). **g**) WT [aTB003], Δ*facZ* [aTB251], and Δ*facZ* cells expressing untagged *facZ* [aTB372] or *facZ-mCherry* [aTB373] under the control of the $P_{tet}$ promoter were normalized to $OD_{600} = 1.0$, serially diluted, and spotted onto TSA agar plates with aTC (50 ng/mL) and with or without PC190723 (100 ng/mL) as indicated.

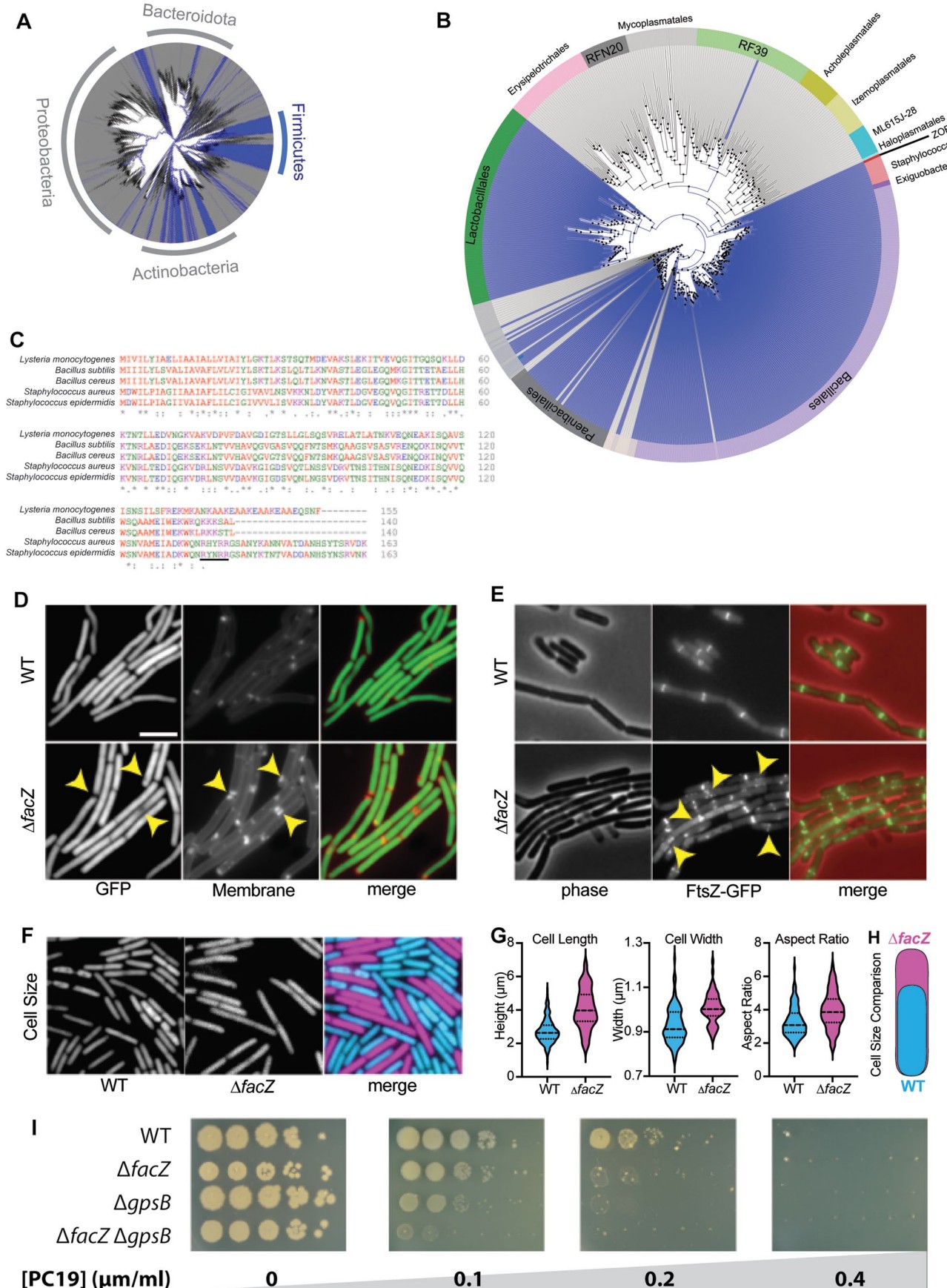

**Extended Data Fig. 5 | See next page for caption.**

**Extended Data Fig. 5 | FacZ is required for normal envelope biogenesis and cell division in *B. subtilis*. a**) Dendrogram highlighting the distribution of FacZ (blue) in a broad range of bacterial species, made in AnnoTree, with large monophyletic phyla labeled. **b**) Dendrogram highlighting the distribution of FacZ orthologs within all monophyletic Firmicute species annotated in AnnoTree, with individual lines representing genera and major families labeled on the periphery of the dendrogram. **c**) Alignment of FacZ orthologs from selected Firmicutes, with proposed GpsB-interaction motif underlined. Colors indicate residue side-chain properties as standard for ClustalOmega alignments (red = small hydrophobic, blue = acidic, magenta = basic, green = hydroxyl/sulfhydryl/amin, gray = unusual/imino). Asterisk (*) indicates single fully-conserved residue; colon (:) indicates strong similarity. **d-g**) All *B. subtilis* microscopy was performed on exponentially-growing derivatives of strain PY79 imaged on agarose pads. All images were scaled identically (scale bar = 4 μm). D) WT [bDR2789] and Δ*facZ* [bTB039] cells that constitutively express cytoplasmic GFP were labeled with the membrane dye Nile Red. Yellow arrowheads highlight membrane invaginations (membrane, middle) that correspond to local

depletions of cytoplasmic GFP (left). These phenotypes are similar to those associated with Δ*facZ* in *S. aureus* (Fig. 3). E) WT [bDR2229] and Δ*facZ* [bTB018] cells expressing FtsZ-GFP from an ectopic locus. Aberrant FtsZ structures are highlighted with yellow arrowheads. F) WT [bDR2637] cells (false-colored cyan) expressing RFP and Δ*facZ* [bTB040] cells (false-colored magenta) expressing BFP were co-cultured and imaged on the same pads, allowing for direct comparison of cell size. G) Cells were segmented using cytoplasmic fluorescence in MicrobeJ to compare cell morphology. Violin plots show cell length, cell width, and aspect ratio of WT and Δ*facZ* cells from a representative experiment (WT, n = 202 cells; Δ*facZ*, n = 68 cells). Dotted lines show median and solid lines show quartiles. **h**) Cartoon depicting the morphological differences between a typical WT and Δ*facZ* cell using the median parameters determined in this analysis (WT length = 2.86 μm, width = 0.916 μm; Δ*facZ* length = 3.97 μm, width = 1.00 μm). **i**) Spot titer of *B. subtilis* strains (WT [bDR2660], ΔfacZ [bTB039], Δ*gpsB* [bTB044], and Δ*facZ* Δ*gpsB* [bTB041]) were grown into exponential phase as described, serially diluted, and plated on a range of PC190723 concentrations.

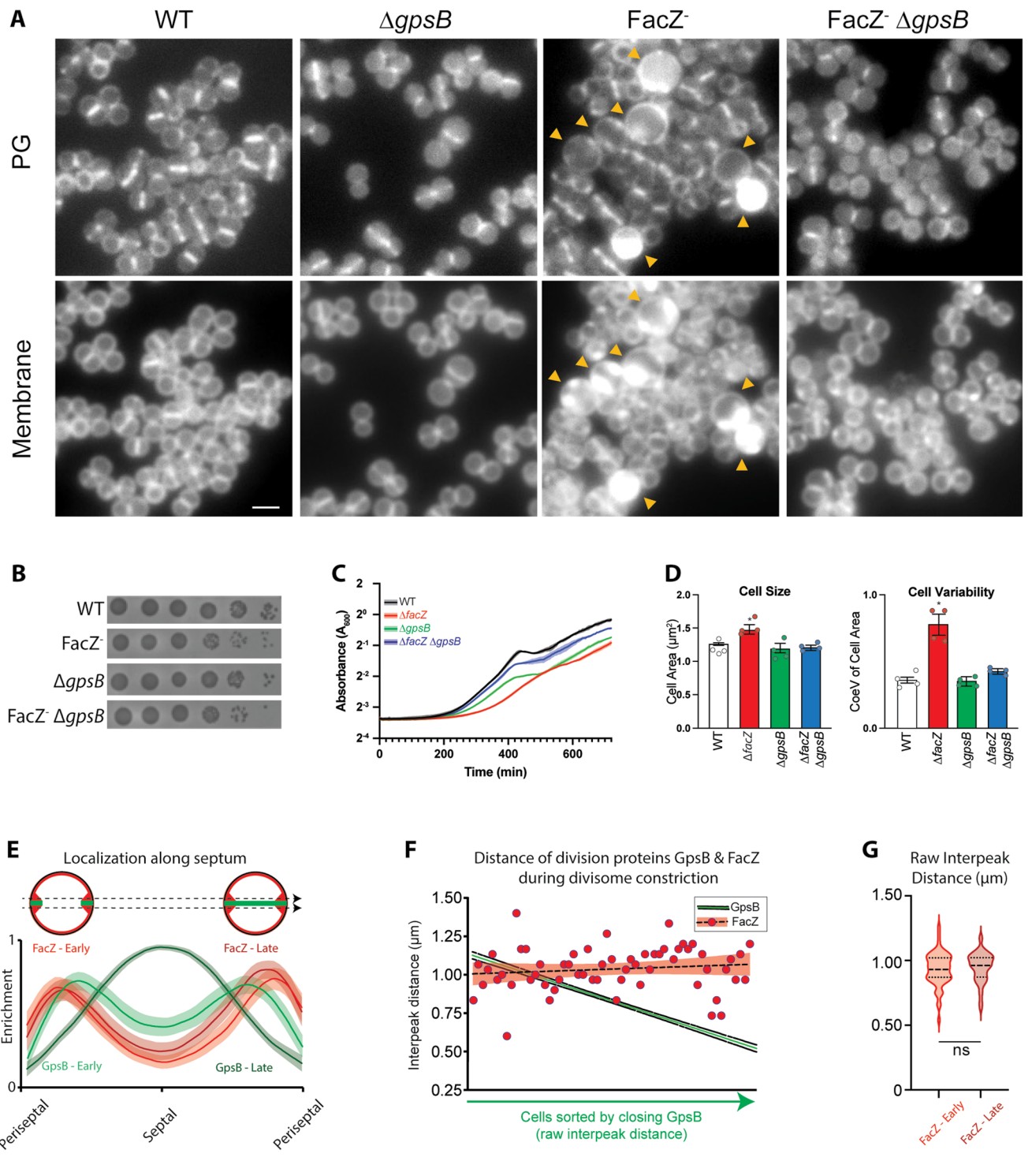

**H** Δ*gpsB* partially rescues FacZ-EzrA synthetic lethality

**I** Δ*facZ* mutants are sensitized to gpsB overexpression

**Extended Data Fig. 6 | See next page for caption.**

**Extended Data Fig. 6 | Δ*gpsB* corrects the morphological defects of Δ*facZ* mutants. a**) Representative images of WT (aTB003), Δ*gpsB* (aTB525), FacZ⁻ (aTB372, *ΔfacZ* P*tet*-*facZ*), and FacZ⁻ Δ*gpsB* (aTB540, *ΔfacZ* P*tet*-*facZ* Δ*gpsB*) grown into mid-log phase without induction of *facZ*, and pulse-labeled with sBADA to label active PG insertion and Nile Red to label cell membranes (see Fig. 5e). Inactivation of *gpsB* has minimal impact on cell morphology or localization of envelope probes. Depletion of FacZ causes characteristic envelope and morphological defects (yellow arrowheads), which are rescued by inactivation of *gpsB* (bar = 2 μm). **b**) Spot titers of the same strains imaged in panel A confirms that depletion of FacZ and/or inactivation of *gpsB* has negligible impact on cell viability. **c**) Growth curves of WT (aTB003), Δ*facZ* [aTB251], Δ*gpsB* [aTB492], and Δ*facZ* Δ*gpsB* [aTB497], grown at 30 C. Values are means of six biological replicates (error bars = 95% CI). Inactivation of FacZ or GpsB causes minor growth defects, while the double mutant is largely restored to WT growth rates. **d**) Median cell size and coefficient of variance of cell size (CoeV) of the indicated strains from replicated experiments (see pooled data in Fig. 5f). Graphs display mean values from four biological replicates, with individual replicate values included as circles, and bars showing standard error. Inactivation of *gpsB* alone has no significant impact on size or variability, but rescues defects caused by depletion of FacZ (Student's *t*-test, p = 0.01). **e**–**g**) Cells were imaged to measure the localization of FacZ and GpsB in different stages of cell division (see Fig. 6a, b). E) Graphs showing the fluorescence intensity of FacZ-mCherry (light and dark red) and GpsB-mNeon (light and dark green) along lines parallel to the septum. Cells were separated into two groups based on the GpsB-mNeon signal: cells with discontinuous GpsB foci were considered early-division cells (top left) whereas cells with a continuous GpsB-mNeon band at the septum were considered late-division cells (top right). (n ≥ 50 cells for each group). F) Graph showing the linear regression of the inter-peak distance of FacZ (red) and GpsB (green) in dividing cells (error bars = 95% CI). Cells were sorted by decreasing inter-peak distance of GpsB, so that measurements are pairwise for individual cells. FacZ and GpsB overlap in early division, but FacZ is left behind at the periseptum as GpsB departs with the closing divisome. G) Violin plots of raw interpeak distance of FacZ in early and late division show no significant change in FacZ localization between early and late division (n = 50 cells). H) Spot titers testing the synthetic interactions of *facZ*, *gpsB*, and *ezrA*. FacZ [aTB378] and GpsB [aTB663] were inactivated separately and together [aTB679] in an EzrA depletion strain [aTB264], and spot titers prepared with or without aTC as in Fig. 4b. Inactivation of *facZ* and *ezrA* is synthetically lethal, while inactivation of *gpsB* and *ezrA* is not. Inactivation of *gpsB* somewhat, but not completely, restores growth in a Δ*facZ* EzrA⁻ strain. I) A plasmid containing P*tet*-*gpsB* was integrated into WT [aTB632] or Δ*facZ S. aureus* [aTB639], and cell viability was assayed by spot titer alongside their parental strains lacking P*tet*-*gpsB* [aTB003 & aTB251] as described (see Methods, Fig. 4b).

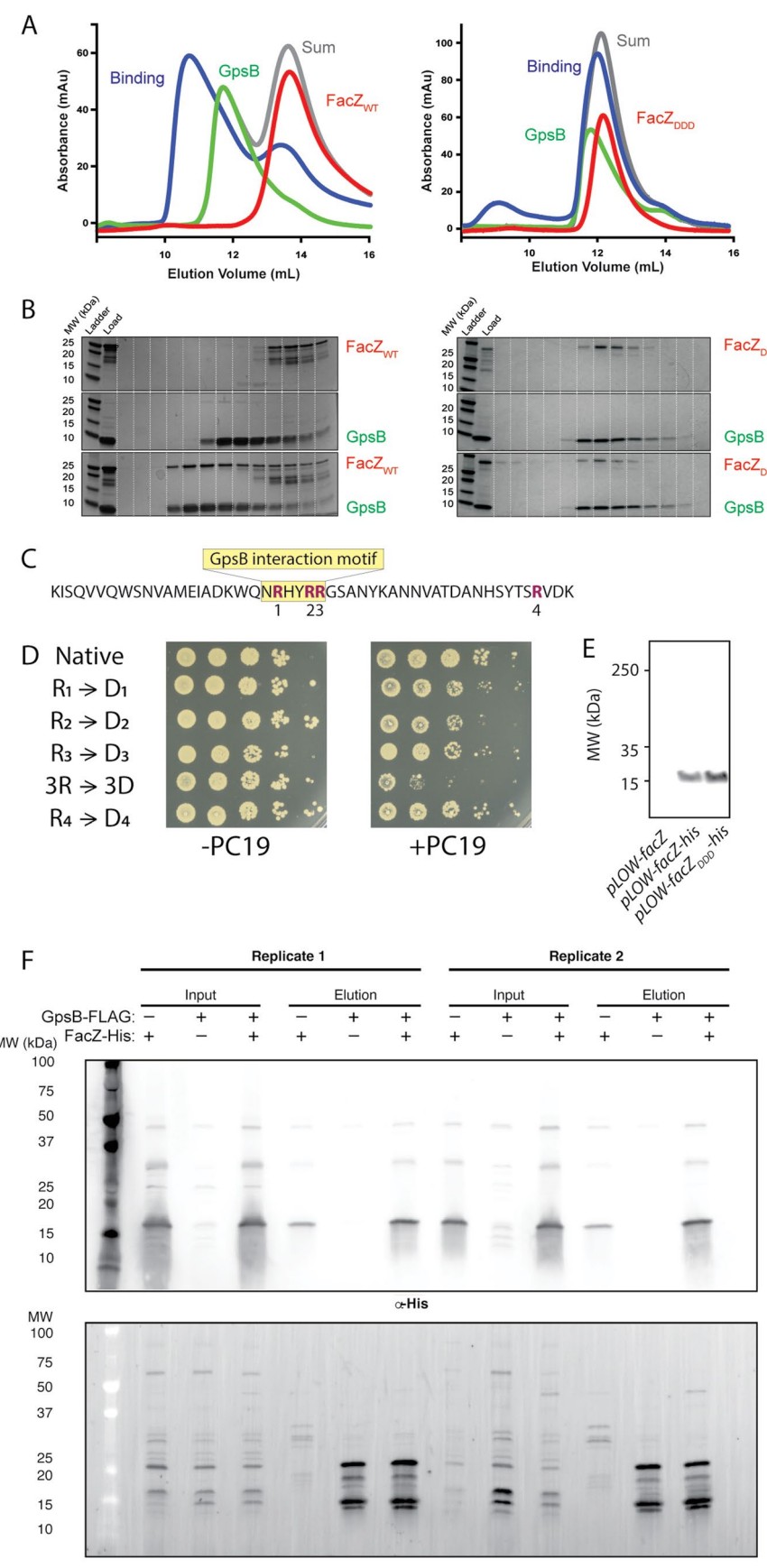

**Extended Data Fig. 7 | See next page for caption.**

**Extended Data Fig. 7 | A C-terminal motif in FacZ is required for function and interaction with GpsB. a, b**) Gel filtration assays testing FacZ-GpsB interactions. Left: Binding assays performed with SUMO-3x-FacZ (127–146) containing the native binding motif (NRHYRR); Right: FacZ Binding assays performed with SUMO-3x-FacZ (127–146) containing an altered motif (NDHYDD). A) Gel filtration profiles of GpsB (1–75) alone (green), SUMO-3x-FacZ (127–146) alone (red), or a mixture of the two proteins (binding, blue). The sum of GpsB (1–75) alone (green) and SUMO-3x-FacZ (127-146) is shown in gray for display purposes. Elution profiles are representative of 6 runs. B) Representative SDS-PAGE gels of fractions from the size-exclusion column from runs with FacZ alone (top), GpsB alone (middle), and a mixture of both proteins (bottom). **c**) Diagram of FacZ C-terminus with proposed GpsB-interaction motif (yellow box) and the location of four arginine residues (bolded red R). **d**) Various R → D FacZ alleles were expressed from $P_{lac}$ on a pLOW vector in a Δ*facZ* background of strain RN4220, and then plated with or without PC19 as indicated. Disruption of R1 [aTB568], R2 [aTB570], or R3 [aTB572] alone had little impact on growth on PC19, while inactivation of all three together (3R → 3D) [aTB565] did not enable growth on PC19. Disruption of R4 [aTB574], outside of the proposed binding motif, had no impact on growth on PC19. **e**) Western blot of native FacZ [aTB347], FacZ-6x-His [aTB349], and FacZ$_{DDD}$ (3R → 3D) [aTB651] expressed from a pLOW vector. **f**) Full blots of FacZ-His and GpsB-FLAG pulldowns from two representative replicates (see Fig. 6e, f).

# Reporting Summary

## Statistics

For all statistical analyses, confirm that the following items are present in the figure legend, table legend, main text, or Methods section.

| n/a | Confirmed | |
|---|---|---|
| ☐ | ☒ | The exact sample size (*n*) for each experimental group/condition, given as a discrete number and unit of measurement |
| ☐ | ☒ | A statement on whether measurements were taken from distinct samples or whether the same sample was measured repeatedly |
| ☐ | ☒ | The statistical test(s) used AND whether they are one- or two-sided *Only common tests should be described solely by name; describe more complex techniques in the Methods section.* |
| ☒ | ☐ | A description of all covariates tested |
| ☐ | ☒ | A description of any assumptions or corrections, such as tests of normality and adjustment for multiple comparisons |
| ☐ | ☒ | A full description of the statistical parameters including central tendency (e.g. means) or other basic estimates (e.g. regression coefficient) AND variation (e.g. standard deviation) or associated estimates of uncertainty (e.g. confidence intervals) |
| ☐ | ☒ | For null hypothesis testing, the test statistic (e.g. *F*, *t*, *r*) with confidence intervals, effect sizes, degrees of freedom and *P* value noted *Give P values as exact values whenever suitable.* |
| ☒ | ☐ | For Bayesian analysis, information on the choice of priors and Markov chain Monte Carlo settings |
| ☒ | ☐ | For hierarchical and complex designs, identification of the appropriate level for tests and full reporting of outcomes |
| ☒ | ☐ | Estimates of effect sizes (e.g. Cohen's *d*, Pearson's *r*), indicating how they were calculated |

*Our web collection on statistics for biologists contains articles on many of the points above.*

## Software and code

Policy information about availability of computer code

| Data collection | Light Microscopy was collected using Nikon NIS Elements 5.1 (referenced in Methods). |
|---|---|
| Data analysis | Code was made available through GitHub |

For manuscripts utilizing custom algorithms or software that are central to the research but not yet described in published literature, software must be made available to editors and reviewers. We strongly encourage code deposition in a community repository (e.g. GitHub). See the Nature Portfolio guidelines for submitting code & software for further information.

## Data

Policy information about availability of data

All manuscripts must include a data availability statement. This statement should provide the following information, where applicable:
- Accession codes, unique identifiers, or web links for publicly available datasets
- A description of any restrictions on data availability
- For clinical datasets or third party data, please ensure that the statement adheres to our policy

All raw sequencing data used to generate graphs has been made available through NCBI SRA, with appropriate links and accession information included in the final manuscript. Key images representing general trends are published in the manuscript. Additional fields of view and substantiating data are available upon request.

## Research involving human participants, their data, or biological material

Policy information about studies with human participants or human data. See also policy information about sex, gender (identity/presentation), and sexual orientation and race, ethnicity and racism.

| | |
|---|---|
| Reporting on sex and gender | *Use the terms sex (biological attribute) and gender (shaped by social and cultural circumstances) carefully in order to avoid confusing both terms. Indicate if findings apply to only one sex or gender; describe whether sex and gender were considered in study design; whether sex and/or gender was determined based on self-reporting or assigned and methods used. Provide in the source data disaggregated sex and gender data, where this information has been collected, and if consent has been obtained for sharing of individual-level data; provide overall numbers in this Reporting Summary. Please state if this information has not been collected. Report sex- and gender-based analyses where performed, justify reasons for lack of sex- and gender-based analysis.* |
| Reporting on race, ethnicity, or other socially relevant groupings | *Please specify the socially constructed or socially relevant categorization variable(s) used in your manuscript and explain why they were used. Please note that such variables should not be used as proxies for other socially constructed/relevant variables (for example, race or ethnicity should not be used as a proxy for socioeconomic status). Provide clear definitions of the relevant terms used, how they were provided (by the participants/respondents, the researchers, or third parties), and the method(s) used to classify people into the different categories (e.g. self-report, census or administrative data, social media data, etc.) Please provide details about how you controlled for confounding variables in your analyses.* |
| Population characteristics | *Describe the covariate-relevant population characteristics of the human research participants (e.g. age, genotypic information, past and current diagnosis and treatment categories). If you filled out the behavioural & social sciences study design questions and have nothing to add here, write "See above."* |
| Recruitment | *Describe how participants were recruited. Outline any potential self-selection bias or other biases that may be present and how these are likely to impact results.* |
| Ethics oversight | *Identify the organization(s) that approved the study protocol.* |

Note that full information on the approval of the study protocol must also be provided in the manuscript.

# Field-specific reporting

Please select the one below that is the best fit for your research. If you are not sure, read the appropriate sections before making your selection.

☒ Life sciences    ☐ Behavioural & social sciences    ☐ Ecological, evolutionary & environmental sciences

For a reference copy of the document with all sections, see nature.com/documents/nr-reporting-summary-flat.pdf

# Life sciences study design

All studies must disclose on these points even when the disclosure is negative.

| | |
|---|---|
| Sample size | Sample sizes were generally not calculated, as large numbers of cells were included (1,000,000 cells for flow cytometry sorting, 50,000 cells for flow analysis without sorting, >500 cells measured for most global cellular-level analysis). When smaller samples were measured for manually-measured phenomena such as counting of cell foci and subcellular localization, the results were representative of replicated studies, and statistically significant differences were observed. |
| Data exclusions | No data were excluded from any analyses. |
| Replication | All microscopy experiments were repeated at least three times. Analyses either represent the pooled data, the mean of multiple experiments, or a representative experiment, as described in figure legends and methods. |
| Randomization | N/A; after data collection, all measurements and analyses were performed identically over all conditions. |
| Blinding | In general, blinding was not possible, as phenotypes of FacZ mutants are very pronounced and fundamentally different. Image processing and analysis procedures were carried out mostly computer-based using unbiased automated procedures, and identical settings were applied wherever possible. |

# Reporting for specific materials, systems and methods

We require information from authors about some types of materials, experimental systems and methods used in many studies. Here, indicate whether each material, system or method listed is relevant to your study. If you are not sure if a list item applies to your research, read the appropriate section before selecting a response.

## Materials & experimental systems

| n/a | Involved in the study |
|---|---|
| ☒ | Antibodies |
| ☒ | Eukaryotic cell lines |
| ☒ | Palaeontology and archaeology |
| ☒ | Animals and other organisms |
| ☒ | Clinical data |
| ☒ | Dual use research of concern |
| ☒ | Plants |

## Methods

| n/a | Involved in the study |
|---|---|
| ☒ | ChIP-seq |
| ☐ ☒ | Flow cytometry |
| ☒ | MRI-based neuroimaging |

# Flow Cytometry

## Plots

Confirm that:

☒ The axis labels state the marker and fluorochrome used (e.g. CD4-FITC).

☒ The axis scales are clearly visible. Include numbers along axes only for bottom left plot of group (a 'group' is an analysis of identical markers).

☒ All plots are contour plots with outliers or pseudocolor plots.

☒ A numerical value for number of cells or percentage (with statistics) is provided.

## Methodology

| | |
|---|---|
| Sample preparation | Cells were grown in liquid culture of TSB with aeration, as described in the text. |
| Instrument | Astrios FACS (Beckman Coulter) |
| Software | FlowJo |
| Cell population abundance | All cells were sorted from highly abundant cell cultures containing many billions of cells, and the first 1,000,000 cells that passed through the gate (see text, and below) were sorted out. |
| Gating strategy | Gating strategy was standardized to WT for "normal" phenotypes and a Δatl mutant for "defective" phenotypes, as displayed in the figures and described in the text. |

☒ Tick this box to confirm that a figure exemplifying the gating strategy is provided in the Supplementary Information.

