## [Peer Review File · Nature Microbiology]

Peer Review Information

Journal: Nature Microbiology

Manuscript Title: Identification of FacZ as a division site placement factor in *Staphylococcus aureus*

Corresponding author name(s): Dr Thomas Bernhardt

Reviewer Comments & Decisions:

Decision Letter, initial version:

Message: 5th June 2023

Dear Tom,

Thank you for your patience while your manuscript "Identification of FacZ as a division site placement factor in *Staphylococcus aureus*" was under peer-review at Nature Microbiology. It has now been seen by 4 referees, whose expertise and comments you will find at the end of this email. Although they find your work of some potential interest, they have raised a number of concerns that will need to be addressed before we can consider publication of the work in Nature Microbiology.

In particular, the referees have concerns regarding the level of mechanistic insights provided, as well as regarding the level of strength of the data for some of the conclusions reached, and have technical concerns. Referee #1 feels that the characterization of FacZ is rather preliminary, and that the main claim (FacZ is a division site placement factor) is only partially supported by the data. Referee #2 has concerns that the data does not really show that FacZ regulates division site placement. Referee #3 suggests an experiment to better prove direct binding between FacZ and GpsB. And referee #4 is concerned by the insufficient molecular mechanistic insights provided, and suggests some additional experiments. Editorially, we would require you to address these concerns.

Should further experimental data allow you to address these criticisms, we would be happy to look at a revised manuscript.

2Please include a data availability statement as a separate section after Methods but before references, under the heading "Data Availability". This section should inform readers about the availability of the data used to support the conclusions of your study. This information includes accession codes to public repositories (data banks for protein, DNA or RNA sequences, microarray, proteomics data etc...), references to source data published alongside the paper, unique identifiers such as URLs to data repository entries, or data set DOIs, and any other statement about data availability. At a minimum, you should include the following statement: "The data that support the findings of this study are available from the corresponding author upon request", mentioning any restrictions on availability. If DOIs are provided, we also strongly encourage including these in the Reference list (authors, title, publisher (repository name), identifier, year). For more guidance on how to write this section please see:

<http://www.nature.com/authors/policies/data/data-availability-statements-data-citations.pdf>

* If you have not done so already we suggest that you begin to revise your manuscript so that it conforms to our Article format instructions at <http://www.nature.com/nmicrobiol/info/final-submission>. Refer also to any guidelines provided in this letter.

When submitting the revised version of your manuscript, please pay close attention to our [href="https://www.nature.com/nature-portfolio/editorial-policies/image-integrity">Digital Image Integrity Guidelines. and to the following points below:](https://www.nature.com/nature-portfolio/editorial-policies/image-integrity)

Note: This url links to your confidential homepage and associated information about manuscripts you may have submitted or be reviewing for us. If you wish to forward this e-mail to co-authors, please delete this link to your homepage first.

Nature Microbiology is committed to improving transparency in authorship. As part of our efforts in this direction, we are now requesting that all authors identified as 'corresponding author' on published papers create and link their Open Researcher and Contributor Identifier (ORCID) with their account on the Manuscript Tracking System (MTS), prior to acceptance. This applies to primary research papers only. ORCID helps the scientific community achieve unambiguous attribution of all scholarly contributions. You can create and link your ORCID from the home page of the MTS by clicking on 'Modify my Springer Nature account'. For more information please visit www.springernature.com/orcid.

If you wish to submit a suitably revised manuscript we would hope to receive it within 6 months. If you cannot send it within this time, please let us know. We will be happy to consider your revision, even if a similar study has been accepted for publication at Nature Microbiology or published elsewhere (up to a maximum of 6 months).

Yours sincerely,

Reviewer Expertise:

Referee #1: Cell division/physiology, *S. aureus*
Referee #2: Cell division/physiology, *S. aureus*
Referee #3: Bacterial cell division
Referee #4: Cell division/physiology

Reviewer Comments:

Reviewer #1 (Remarks to the Author):

Review NMICROBIOL-23040978 "Identification of FacZ as a division site placement factor in *Staphylococcus aureus*".
General comment: In this very well-written manuscript a dual cell-sorting phenotypic screen (Tn-seq used to determine enrichment) was used to identify uncharacterized cell division genes in *Staphylococcus aureus* – a commensal bacteria that is a major cause of hospital-acquired infections. One gene was characterized in more detail and is proposed to function as a division site placement factor/coordinator of the Z-ring (the text is inconsistent). Although the manuscript presents some valuable findings, the characterization of FacZ is rather preliminary and the main claim (FacZ is a division site placement factor) is only partially supported by the presented data.

Major points

- Throughout the text it is unclear which background strains were used for the experiments and details of how the *facZ* deletion strain are missing. This is important as Fig 2A reveals that all transposon insertions in the 1855 gene are located at the same site very close to the 3' end of the gene strongly indicating that the *FacZ* proteins could still be expressed and (partly) functional. Alternatively, the phenotypes described could be linked to impaired expression of the downstream gene as Aureowiki (https://aureowiki.med.uni-greifswald.de/SAOUHSC_01855) shows that the 1855 gene is the first gene in a bicistronic operon. It is crucial to address the potential polar effects of the Tn-insertion and the insertion of the spec-cassette in the 1855 gene on expression of this downstream gene.
- *FacZ* (Firmicute-associated coordinator of Z-ring) is proposed to function as a division site placement factor/coordinator of the Z-ring, however, there is no strong data supporting either of these two functions. What has been shown is that:
- Fig. 3 and S.3 + line 161: "...its absence results in the formation of spurious invaginations of the envelope that we suspect represent aberrant division attempt". I tend to agree with this conclusion but, importantly, this also rules out a septum placement defect as the division attempts take place at mid-cell both in *S. aureus* and *Bacillus*. Overall, the microscopic data presented is descriptive and very little quantification has been performed, and it is difficult to differentiate between direct and indirect effects or whether *FacZ* affects membrane biogenesis, or septal PG-synthesis.
- Line 193 (localization of *FacZ*): "These intensity profiles ...mirror profiles obtained for cells labeled with the membrane dye, Nile Red". This indicates that the small *FacZ* enrichment at the periseptal site is due to the double membrane, implying that *FacZ* is uniformly distributed in the cytoplasm and arguing against a septal function of the protein. Please provide quantitative data to prove *FacZ* enrichment at the periseptal site.
- Tn-Seq identified the cell division initiator protein, *EzrA* as a synthetic lethal partner for *FacZ*. At first, I thought that this was a good indication that the function of *FacZ* is related to cell division but Table S6 revealed that a large number proteins with diverse functions is synthetically lethal with the *facZ* deletion again indicating that *FacZ* could have a more general role in the cell. In the current manuscript there is no mentioning or discussion of the proteins listed in Table S6 and the interpretation of this data seems very biased. Please, provide a thorough discussion of this data.
- Finally, the data showing that the *facZ* mutant is hypersensitive to an inhibitor of *FtsZ* GTPase activity again indicates a link between *FacZ* and *FtsZ* but it remains unclear if *FacZ* interacts directly with *FtsZ* or if cells become hypersensitive to PC for indirect reasons.
- In conclusion, I do not find that the presented data justify the *FacZ* name (although I have used the name throughout this review!).

Minor things

- The use of the UV-mutagenized and aberrant RN4220 as the model strain is problematic as the strain possesses numerous uncharacterized mutations and the limitations of using the RN4220 model strain should be clearly stated in the text.
- The two criteria used in the cell-sorting (membrane damage measured as PI-permeability and cell-clumping as observed for a mutant lacking the *Atl* cell wall hydrolase involved in daughter cells splitting) seem more directed against finding factors involved in controlling autolytic activity and are not well-aligned with the aim of finding new factors involved in cell envelope assembly? Accordingly, the top-scores from the screening are the autolytic enzymes *atl* and *sagB* (line 133). The autolytic bias of screen needs to be discussed throughout the

text - in the current introduction there is a focus on divisive placement that seems odd.

- In supplementary table 4, the most highly enriched genes in both screens are listed according to gene numbers while gene names are used in the graphical presentation. To get the full information from this data, gene names and putative function should be added to the supplementary tables throughout.

- Line 150-151 describing the TEM pictures in Fig. 3A: It is not clear what the author interpret as the "aberrant accumulation of envelope" in the TEM image of the *facZ* mutant? Given the heterogeneity of the described phenotypes, a TEM picture showing changes at the population level is warranted.

Reviewer #2 (Remarks to the Author):

The manuscript by Bartlett et al. reports a genetic screen in *Staphylococcus aureus* to identify genes involved in envelope biogenesis and cell division. The authors identified multiple candidate genes and focused on one gene of previously unknown function, which they named "facZ", which was previously also identified in *B. subtilis* as a gene required for efficient sporulation. The authors propose that FacZ antagonizes the cell division protein GpsB to reinforce proper Z-ring assembly in Firmicutes. In general, the experiments were of high quality, and the paper was well-written and should be accessible to a broad audience of microbiologists. I have some largely cosmetic suggestions that the authors may or may not choose to incorporate.

Major comments

1. Line 158, Fig. S3D-E. Is there a reason the complementation of the *facZ* deletion was performed with an inducible promoter? If complementation with the native promoter at an ectopic locus did not support complementation, consider reporting this observation.

2. Fig. 4C-D. The image analysis of the periseptal localization of FacZ appears to be performed using diffraction-limited microscopy. Have the authors quantified this localization pattern using a super-resolution method such as SIM (which they employed in Fig. 2)? If so, the separation of the FacZ signal on either side of the division septum may be more pronounced.

3. Fig. 4. The FacZ-mCherry fusion was produced under control of an inducible promoter. Is there a concern that the expression levels of this construct may be different than WT expression level, which may contribute to the peripheral localization pattern? Can the authors compare the relative expression level of the fusion construct compared to the expression level of native FacZ? Is FacZ-mCherry visible when produced under its native promoter?

4. Fig. 6C, lines 238-239. Does the deletion of other cell division genes, (such as *ezrA*, which the authors used for comparative purposes), also result in mis-localization of GpsB, similar to the mis-localization of FtsZ shown in Fig. S3C? The authors are invoking a direct effect of the deletion of *facZ* in mis-localizing GpsB (which reportedly associates with FtsZ), but I wonder if a general defect in Z-ring assembly (similar to deletion of *ezrA*) also causes a mis-localization of GpsB and other divisive/FtsZ-associated proteins?

65. Fig. 6D-E. The authors invoke an interaction between GpsB and FacZ, but the subcellular localization of the two proteins are almost entirely non-overlapping: GpsB appears to localize at mid-cell with FtsZ, but FacZ is present on the cell periphery and the periseptal region. Can the authors speculate why two proteins that interact robustly in vitro would not largely co-localize in vivo?

6. Lines 249-251, lines 270-271. I am a bit confused by the model as presented. The authors state that FacZ 'prevents aberrant envelope invaginations', but also imply that FacZ only localizes once 'membrane invagination is initiated'. Please clarify. On a more semantic note, although it is clear from the data that FacZ is an important regulator of Z-ring formation, is it proper to refer to a late-divisome protein (which localizes only after cell division initiates) as a 'division site placement factor' in the Title? The abstract also suggests that FacZ can control 'division site placement in *S. aureus* and other Firmicutes', but it appears that *B. subtilis* (Fig. S5B) finds mid-cell efficiently even in the absence of FacZ, arguing against a role in division site placement, per se. Consider softening the conclusion and the title with respect to 'placement', especially since other important cell division factors like EzrA also show a similar Z-ring assembly defect when removed and are not referred to as 'placement factors'.

Minor comment

1. Line 273. The authors describe the junction of the septal and peripheral membranes as 'positive' Gaussian curvature, but since the principal normal curvatures on the cytoplasmic face of the membrane curve in opposite directions, I believe this should be 'negative' Gaussian curvature (as measured from inside the cell, where FacZ localizes), correct?

Reviewer #3 (Remarks to the Author):

This paper presents innovative variations of cell sorting of transposon-mutagenized cells of *Staphylococcus aureus* to catalog insertions in non-essential genes that affect cell size/clustering and/or membrane permeability. The screening methods are validated and compared. As a control, they turn up known genes in cell envelope synthesis and assembly as well as a number of new genes, which are listed in the figures and supplemental tables. The paper then focuses on one of these new genes, *facZ*, which has substantial effects on cell size, division/clumping, and membrane permeability. The paper goes on to characterize some phenotypes and genetic interactions of *facZ*. Delta-*facZ* mutants have unusual phenotypes of forming clumps of larger cells with membrane and peptidoglycan mislocalized into foci adjacent to nucleoids. These phenotypes are complemented in control experiments. *facZ* homologs are found in some, but not all Firmicutes, and delta-*facZ* mutants of *B. subtilis* also have some defects that parallel those in *S. aureus*. The authors then go on to characterize additional genetic relationships of *facZ* in *S. aureus*. They show that localization of FtsZ is perturbed in a *facZ* mutant in a way reminiscent of in *ezaA* mutants. Fusion to mCherry shows that FacZ localizes around the cell membrane with concentration to the periseptal region of dividing *S. aureus* cells. Tn-seq and follow-up experiments show that EzrA and FacZ are synthetically lethal and that mutants lacking EzrA or FacZ are hypersensitive to the PL19 compound that inhibits FtsZ GTPase in many Gram-positive bacteria. Null mutations in *gpsB*

7suppressed the hypersensitivity of delta-facZ mutants to PC19, and delta-gpsB mutations restored a near-WT cell phenotype to delta-facZ mutants. In addition, FacZ was required for normal septal localization of GpsB. Last, FacZ contains a known binding motif for GpsB, and a repeated version of this binding motif from FacZ bound to a domain of GpsB. Together, these results lead to model where FacZ concentrates to the periseptum and localizes and restricts GpsB activity to the septal region, thereby preventing aberrant GpsB activity around *S. aureus* cells.

This paper presents a useful screen and new candidates with roles in cell surface biogenesis in *S. aureus*. The screen is convincingly validated, and the new candidates will be the subject for new studies. In addition, the study of FacZ function presents many interesting new results that establish FacZ as an important new division protein in *S. aureus* through positioning of the GpsB regulator. The paper is well written and convincing for the most part. Overall, the data is of high quality and clearly presented. However, there are a few topics that require greater context and explanation. Several simple experiments would strengthen some of the main conclusions, and the biochemical experiments are not complete enough. Some suggestions are included below that would improve this very interesting study.

1. Line 42-43, Fig. S4, and Discussion (line 288). There is no evidence yet for “a conserved role” in “other Firmicutes” (but there could be with simple additional experiments in point 2, below). There is a sizable slice of the Firmicute pie missing in Fig. S4. Please specify which Firmicute groups do not have FacZ, but still contain GpsB? This would provide context on how conserved this role is. Along this line, line 43 should be changed to “and certain other Firmicutes.” It would help to redraw Fig. S4 larger and with more detail to indicate the Firmicute groups with and without FacZ and whether any also lack GpsB. It is also curious that Actinobacteria also contain FacZ. Do they also contain a GpsB homolog? Please consider mentioning this topic should very briefly in the Discussion.
2. To generalize the mechanism more, does delta-gpsB reverse the delta-facZ phenotypes in *B. subtilis* (Fig. S5). Along the same line, does FacZ in *B. subtilis* and the other selected Firmicutes in Fig. S4 contain a conserved GpsB binding site? Also, is a delta-facZ *B. subtilis* mutant hypersensitive to PC 19 and synthetically lethal with delta-ezrA. Results from these simple experiments would be highly informative that the mechanism in *S. aureus* extends to *B. subtilis*.
3. One other point of context that should be mentioned in the Discussion is that FtsZ from *S. aureus* is unusual in that it contains a special C-terminal tail that binds to the N-terminal domain of GpsB (Sacco et al. in references). This tail is not found in FtsZ homologs from some other species. Therefore, there would seem to be a competition for GpsB binding between FtsZ and FacZ (and PBP4) that does not occur in other species, including *B. subtilis*. This competition could play into the FacZ mechanism reported here and may be unique to Firmicute species like *S. aureus* that have this type FtsZ-GpsB interaction.
4. Line 133 and line 50 in Supplemental. It's stated that the PCA correlated genes have well known functions. It would help readers outside of *S. aureus* to list gene functions, possibly in Fig. S1, where there is room for an addition.
5. Line 144. What are the growth properties of the delta-facZ mutant in *S. aureus*? Do the

cells grow more slowly and to a lower final OD, and do the cells remain viable in exponential and stationary phase? Since this is the first report of this gene, this information is of interest. Also, the micrographs seem to indicate a considerable heterogeneity of cell sizes of the delta-facZ mutant that is not fully reflected in the graphs of cell areas. Please clarify this point briefly in the text.

6. Line 182. You might mention that Fig. S6 shows that the FacZ-mCherry fusion is functional, which is an important point.

7. Figure 2A. Why is there only one enriched Tn insertion in the C-terminus of facZ? Is there only one TA in facZ or could this mean that the facZ phenotypes only appear when the C-terminus is inactivated. Please clarify.

8. Line 204. Table S6 seems to show that there are other genes that are synthetically lethal with FacZ. Is this correct, and if so, it should be commented on. Also, based on the previous publications about EzrA function in *S. aureus*, is FtsZ-ring formation the only reason underlying the synthetic lethal relationship between FacZ and EzrA. This part of the story seems to drop off. Does delta-gpsB also suppress the hypersensitivity of delta-ezrA to PC 19 or are delta-gpsB and delta-ezrA synthetically lethal in *S. aureus*? Please expand slightly.

9. Line 219. Were other suppressors of PC19 hypersensitivity of delta-facZ identified in genes other than gpsB? If so, which genes were they? The identity of other suppressors should be listed and could be highly relevant to the model.

10. Line 228. It is remarkable that delta-gpsB seems to have minimal phenotypes (Fig. S7). This point might be emphasized more. The results of delta-gpsB alleviating the delta-facZ defects is very nice, as is the demonstration that GpsB localization is dependent on FacZ in *S. aureus*.

11. Line 241 and Fig. 6D. The biochemical experiments are incomplete. An appropriate experiment was done showing that three tandem repeated copies of the putative GpsB binding site from FacZ fused to SUMO binds to the N-terminal domain of GpsB. However, Fig. 6D should be relabeled, because "FacZ" was not really used. Does a single copy of the binding site, as in the native protein (Fig. S6), fused to SUMO also bind to the GpsB domain? Most definitely, does the purified cytoplasmic domain of FacZ bind to GpsB? These experiments should be tried and included, if they can be done, to really indicate direct binding between FacZ and GpsB. The GpsB binding sites might be indicated on the predicted models of FacZ in Fig. 6B and 6C. From the numbering in the text, it seems like the GpsB binding motif is in the prongs of the "anchor" structure.

12. Line 264. Please qualify to: "interact directly with FtsZ in *S. aureus* and to promote...", because GpsB binding to FtsZ has not been detected in other Gram-positive species (please see point 3, above).

Reviewer #4 (Remarks to the Author):

9This paper does a nice job of finding and describing a novel nonessential protein that influences cell division. A molecular mechanism for how this occurs is unclear but it appears to somehow restrict Z ring number or orientation.

This paper uses flow cytometry to find genes in *S. aureus* involved in cell morphology/envelope integrity. Using this approach, they find expected genes as well as genes of unknown function. They pick one of these, designated FacZ, and examine it in some detail because of the curious envelope defects/morphology that includes internal membrane deposition that is accompanied with cell wall deposition and connected to the peripheral envelope. A knockout of FacZ leads to altered morphology which they trace back to spurious/multiple Z rings. Interestingly, a knockout of GpsB, which has little phenotype, appears to almost completely reverse the FacZ phenotype. Thus, deletion of two nonessential genes cancel each other out leading one to wonder what they are doing.

The first part of the paper describes and validates the two screens that were used – mutants with increased light scattering and those with increased uptake of a dead cell dye (Figs. 1 & 2 and S1 and S2 and Tables S1-6). The results indicate that the screens yield somewhat overlapping genes. Since, they are screening for underrepresented genes from random transposon libraries that give the above phenotypes, they are looking for nonessential genes that affect morphology/envelope integrity. The description of the screen and the reporting of the results are very thorough.

Further study of FacZ shows it's a peripheral membrane protein not enriched at the septum. However, FacZ is connected to division since it is synthetic lethal with EzrA and like an EzrA mutant is sensitive to an FtsZ inhibitor. Curiously, the authors find that this sensitivity is alleviated by inactivating GpsB. Whereas GpsB appears to be mostly at the septum in WT cells, in a FacZ mutant it is associated with the unusual membrane depositions that led the authors to study FacZ in the first place. If GpsB is considered a septal protein then this argues that these depositions are septal material. This should be confirmed by examining the localization of an essential protein to verify that it is septal synthesis – any of FtsQ, L, B, W or I would suffice.

Other comments:

Fig. 3A. How come the FacZ mutant lacks the red halo from the membrane stain?

Fig. S4. I assume the blue indicates species that have the gene. This means more species lack the gene than have it?

Lines 190-202. Description of FacZ localization. In line 201 it is concluded that "FacZ is enriched at midcell". However, on lines 193-4 it is stated that the intensity profile is "as expected for a membrane protein". I think I understand what the authors are getting at but it is a bit confusing.

Fig. S7D. What is the size of FacZ based on elution volume? Monomer or multimer? Only full length FacZ elutes with GpsB – the breakdown products do not. Is it known if these are missing the N or C terminus?

10Author Rebuttal to Initial comments

Responses to reviewer comments Reviewer #1

(Remarks to the Author):

Major points

1) Throughout the text it is unclear which background strains were used for the experiments and details of how the *facZ* deletion strain are missing. This is important as Fig 2A reveals that all transposon insertions in the 1855 gene are located at the same site very close to the 3' end of the gene strongly indicating that the FacZ proteins could still be expressed and (partly) functional. Alternatively, the phenotypes described could be linked to impaired expression of the downstream gene as Aureowiki shows that the 1855 gene is the first gene in a bicistronic operon. It is crucial to address the potential polar effects of the Tn-insertion and the insertion of the spec-cassette in the 1855 gene on expression of this downstream gene.

Response: We appreciate the reviewer's concern here. Where appropriate, we now indicate the background strains used for the experiments in the main text and figures legends. The initial screen was performed in the RN4220 background. Subsequent work was almost entirely performed in HG003. The morphological and growth phenotypes caused by the *facZ* deletion allele ($\Delta facZ::spec^R$) are fully complemented by the expression of *facZ* from a plasmid. This result rules out polar effects on the downstream gene playing a significant role in the phenotypes we observe.

2) FacZ (Firmicute-associated coordinator of Z-ring) is proposed to function as a division site placement factor/coordinator of the Z-ring, however, there is no strong data supporting either of these two functions. What has been shown is that:

a) Fig. 3 and S.3 + line 161: "...its absence results in the formation of spurious

invaginations of the envelope that we suspect represent aberrant division attempt". I tend to agree with this conclusion but, importantly, this also rules out a septum placement defect as the division attempts take place at mid-cell both in *S. aureus* and *Bacillus*. Overall, the microscopic data presented is descriptive and very little quantification has been performed, and it is difficult to differentiate between direct and indirect effects or whether FacZ affects membrane biogenesis, or septal PG-synthesis.

Response: We agree with the reviewer that it can be difficult to rule out direct effects from indirect ones based on mutant phenotypes. However, we do not agree that the observation of midcell division events in some cells rules out a septum placement defect. It is common for mutants lacking division placement factors to display a heterogenous phenotype with some cells dividing at midcell and others dividing aberrantly. For example, mutants of *E. coli* or *B. subtilis* lacking the Min system, the classic and most well-studied division regulatory system, often divide correctly at midcell. Such a phenotype is not unexpected due to there being several partially redundant division regulators working together in cells. Thus, the partially penetrant phenotype of $\Delta facZ$ is not surprising and does not rule out a role for FacZ in septum placement.

While it can be difficult to determine which effects are direct and which are indirect, we show that the phenotypes of $\Delta facZ$ mutants can be reversed by inactivation of GpsB. Furthermore, we show that FacZ and GpsB interact directly *in vitro* and that this interaction is required for FacZ function. Thus, our results make a strong case that FacZ functions as a direct regulator of GpsB to control septum formation and prevent aberrant septation events.

FacZ mutants have a significant defect in Z-ring placement and form many extra FtsZ structures. This observation is why we initially chose the name FacZ (Firmicute associated coordinator of Z-rings). However, we can see how this name may unintentionally imply that FacZ is a direct regulator of FtsZ, which is unlikely to be the case since all experiments point towards it controlling GpsB activity. We have therefore kept the gene designation of *facZ* but changed the descriptor to "Factor preventing extra Z-rings".

Finally, we disagree with the assertion that very little quantification of microscopy data was performed. Almost every imaging experiment in the paper is associated with relevant image analysis and quantification. Nevertheless, we have added

additional quantification of some experiments (see Figure 6B) to clarify that the localization of FacZ and GpsB overlap during the cell cycle during the earliest stages of cell division.

b) Line 193 (localization of FacZ): “These intensity profiles ...mirror profiles obtained for cells labeled with the membrane dye, Nile Red”. This indicates that the small FacZ enrichment at the periseptal site is due to the double membrane, implying that FacZ is uniformly distributed in the cytoplasm and arguing against a

septal function of the protein. Please provide quantitative data to prove FacZ enrichment at the periseptal site.

Response: Given that the periseptal membrane region is diffraction-limited, we repeated the FacZ localization experiments with 3D-SIM. Unfortunately, the FacZ-mCherry signal was not bright enough to yield quality reconstructions. In addition, the septal membranes are close enough that, even with this improved resolution, it was not possible for us to resolve the opposed membranes in the periseptal region. Nevertheless, we have performed the quantification of the data requested by the reviewer by comparing FacZ enrichment at the septum vs the periphery to Nile Red signal at the septum vs the periphery. We found that FacZ periseptal enrichment is 1.18-fold stronger than Nile Red enrichment at the periseptum, indicating that the observed periseptal enrichment cannot be explained merely by the increased abundance of membrane at the periseptum. We have added a line of text to include this observation and a supplemental figure panel (see panel S4F) to illustrate this point. Importantly, our model is that FacZ is an antagonist of ectopic division site selection, and we do not necessarily predict, or intend to invoke, that FacZ is *more abundant* at any part of the septum or periseptum, nor do we think this is necessary for the function of a division regulator (the Min proteins in *E. coli* are not enriched at the septum nor are Noc or SlmA). Our model merely predicts that FacZ is present at the periseptum, where we have shown it to be modestly enriched.

c) Tn-Seq identified the cell division initiator protein, EzrA as a synthetic lethal partner for FacZ. At first, I thought that this was a good indication that the function of FacZ is related to cell division but Table S6 revealed that a large number proteins with diverse functions is synthetically lethal with the facZ deletion again indicating that FacZ could have a more general role in the cell. In the current manuscript there is no mentioning or discussion of the proteins listed in Table S6 and the interpretation of this data seems very biased. Please, provide a thorough discussion of this data.

Response: In our extensive experience with synthetic lethal screens, we often identify many unexpected genetic interactions with a gene of interest, particularly when screening for interactions with genes encoding factors involved in cell division. For example, in our original screens for mutations synthetically lethal with a defective Min system in *E. coli*, the hits included cell shape proteins, ribosomal proteins, factors involved in RNA processing, etc... Similarly, in our screen for mutations synthetic lethal with Δnoc in *S. aureus*, the hits included

14

factors unrelated to division. The reason for the diversity of functions in the “hit list” is not because the Min/Noc systems have a general role in the cell, but instead is because of the many cellular activities that impact division. It is only when division is partially compromised that defects in these cellular activities cause a strong growth defect. The situation with FacZ is likely to be similar to that of the Min/Noc systems. Only some synthetic lethal partners like EzrA readily make sense and are consistent with a division function for FacZ that is strongly

supported by all its other phenotypes (extra Z-rings, spurious invaginations, PC19 sensitivity, GpsB interaction...).

Based on the logic above, we do not think the other hits in the list of synthetic lethal interactors weakens the argument for FacZ playing a role in division. Nevertheless, we now indicate that synthetic lethal partners in factors not directly associated with cell division were also identified in the screen.

d) Finally, the data showing that the *facZ* mutant is hypersensitive to an inhibitor of FtsZ GTPase activity again indicates a link between FacZ and FtsZ but it remains unclear if FacZ interacts directly with FtsZ or if cells become hypersensitive to PC for indirect reasons.

Response: We apologize for any confusion the original text may have caused, but we never meant to imply that FacZ is directly interacting with FtsZ to control its function. Rather, we think it exerts its effect on GpsB. It is GpsB that interacts directly with FtsZ. Our model to explain the $\Delta facZ$ phenotype is that in the absence of FacZ, GpsB is overactive in promoting division through its association with FtsZ, leading to many aberrant septation events. The model is strongly supported by the observation that GpsB inactivation suppresses the phenotypes of $\Delta facZ$ mutants, and that FacZ and GpsB interact directly (as well as prior work showing that GpsB and FtsZ interact directly). We have revised the main text to make this model clearer.

In conclusion, I do not find that the presented data justify the FacZ name (although I have used the name throughout this review!).

Response: As mentioned above, we have kept the name FacZ but changed the descriptor to “Factor preventing extra Z-rings”. We have also clarified the discussion of our model to avoid the misinterpretation that FacZ acts via a direct interaction with FtsZ. We think that the data strongly support the model that FacZ is a regulator of cell division that acts by antagonizing the activity of the highly conserved GpsB protein. This mode of division regulation is novel and its discovery therefore represents an important advance in our understanding of how *S. aureus* controls septum placement. In the resubmission, we emphasize this point in the Summary, which now reads:

"Thus, FacZ is a novel division site placement factor in *S. aureus* with an unconventional mode of action. Rather than acting at the level of FtsZ, it antagonizes the activity of GpsB, an FtsZ-

binding protein in *S. aureus* that forms an interaction hub at the division site connecting several envelope biogenesis factors with the cytokinetic ring.”

Minor things

3) The use of the UV-mutagenized and aberrant RN4220 as the model strain is problematic as the strain possesses numerous uncharacterized mutations and the limitations of using the RN4220 model strain should be clearly stated in the text.Response: We apologize for the confusion. As mentioned above, RN4220 was used for screening and as a cloning intermediate. Virtually all other experiments were conducted in HG003 or USA300. Strain usage has been clarified throughout the manuscript, and is also included in Table S8.

We note that *E. coli* MG1655 was also extensively mutagenized early in its history. It has nevertheless served as a useful model system over the years, as has RN4220. Neither are perfect models, if any such model exists, but they have helped discover many fundamental processes in their respective species and in bacteria in general.

4) The two criteria used in the cell-sorting (membrane damage measured as PI- permeability and cell-clumping as observed for a mutant lacking the Atl cell wall hydrolase involved in daughter cells splitting) seem more directed against finding factors involved in controlling autolytic activity and are not well-aligned with the aim of finding new factors involved in cell envelope assembly? Accordingly, the top-scores from the screening are the autolytic enzymes *atl* and *sagB* (line 133). The autolytic bias of screen needs to be discussed throughout the text - in the current introduction there is a focus on divisome placement that seems odd.

Response: The reviewer is correct that one might expect use of the *atl* mutant for the gating might bias the FACS screens for mutants defective in the autolytic system. Indeed, as the reviewer points out, many of the hits were in cell wall hydrolases involved in cell separation. However, neither an increased PI permeability nor an increased cell size is a specific phenotype for autolytic mutants. The *atl* mutant was just the most convenient control we had at the time to set the gates for the FACS sorting. In the revised text describing the screen, we now explicitly state the objective is to broadly identify cell division and size mutants with the screens, and that the autolytic mutant was just a useful control to test whether mutants with increased size and/or PI-permeability could successfully be sorted using our methods.

5) In supplementary table 4, the most highly enriched genes in both screens are listed according to gene numbers while gene names are used in the graphical presentation. To get the full information from this data, gene names and putative function should be added to the supplementary tables throughout.

Response: We have added a table of the top 50 gene hits for each screen that now includes functional annotations to aid readers in combing through all the hits obtained. We thank the reviewer for prompting us to include this list.

6) Line 150-151 describing the TEM pictures in Fig. 3A: It is not clear what the author interpret as the “aberrant accumulation of envelope” in the TEM image of the *facZ* mutant? Given the heterogeneity of the described phenotypes, a TEM picture showing changes at the population level is warranted.

Response: As suggested by the reviewer, we have added supplemental figure panels (Figure S3G) with full fields of view to better represent the full scope of the $\Delta facZ$

phenotype observed by TEM analysis. The phrase “aberrant accumulation of envelope” refers to the bright spots in the insets in Fig. 3A. This designation is now indicated in the legend. We have also added carets to Fig S3G pointing to cells that have discrete internal bright spots or projections of the cell envelope. Our fluorescence microscopy suggests that these are likely invaginations of cell envelope, as we see similar invaginations by fluorescence microscopy, which are composed of both wall and membrane. Consistent with this inference, these spots have the same coloration and intensity as the cell envelope.

Reviewer #2 (Remarks to the Author):

The manuscript by Bartlett et al. reports a genetic screen in *Staphylococcus aureus* to identify genes involved in envelope biogenesis and cell division. The authors identified multiple candidate genes and focused on one gene of previously unknown function, which they named “facZ”, which was previously also identified in *B. subtilis* as a gene required for efficient sporulation. The authors propose that FacZ antagonizes the cell division protein GpsB to reinforce proper Z-ring assembly in Firmicutes. In general, the experiments were of high quality, and the paper was well-written and should be accessible to a broad audience of microbiologists. I have some largely cosmetic suggestions that the authors may or may not choose to incorporate.

Response: We thank the reviewer for their careful reading of our manuscript and for their helpful feedback. We also appreciate the reviewer’s enthusiasm for the work.

Major comments

1. Line 158, Fig. S3D-E. Is there a reason the complementation of the facZ deletion was performed with an inducible promoter? If complementation with the native promoter at an ectopic locus did not support complementation, consider reporting this observation.

Response: We typically use inducible promoters for complementation studies. The inducibility provides a convenient way to “turn the phenotype on or off” for analysis. We

did not try complementation with *facZ* under control of its native promoter nor do we think it is necessary for any of our conclusions.

2. Fig. 4C-D. The image analysis of the periseptal localization of FacZ appears to be performed using diffraction-limited microscopy. Have the authors quantified this localization pattern using a super-resolution method such as SIM (which they employed in Fig. 2)? If so, the separation of the FacZ signal on either side of the division septum may be more pronounced.

Response: The reviewer raises a good point. We attempted to analyze FacZ-mCherry localization by 3D-SIM, but the signal is not bright enough for high quality imaging. We now present possible modes of FacZ localization at the septum in Fig. S4E. In short, FacZ may be enriched at the highly-curved periseptal junction, or decorate membranes

on either side of this junction (i.e., FacZ may penetrate slightly into the flat septal membranes, and its periseptal enrichment might be due to increased membrane abundance at the periseptal junction). We think the former is more likely, as periseptal enrichment of FacZ is greater than periseptal enrichment of Nile Red by 1.18-fold, but we do not think that either can currently be ruled out.

3. Fig. 4. The FacZ-mCherry fusion was produced under control of an inducible promoter. Is there a concern that the expression levels of this construct may be different than WT expression level, which may contribute to the peripheral localization pattern? Can the authors compare the relative expression level of the fusion construct compared to the expression level of native FacZ? Is FacZ-mCherry visible when produced under its native promoter?

Response: We unfortunately do not have antibodies to native FacZ and thus cannot compare the levels of FacZ-mCherry produced in the localization experiments to those produced from the native locus. Expression of *facZ-mCherry* from the native promoter would not resolve this issue because the tag may affect mRNA and/or protein stability such that there is no guarantee expression levels would be the same as untagged *facZ*. The best we could do was to use the minimum level of inducer needed for the *facZ-mCherry* construct to complement the *facZ* deletion. It is at these levels we see the peripheral signal (see panels S3D-E). We cannot rule out that enrichment of native FacZ at the periseptum might be greater than we observe with this construct, but definitively demonstrating this is not critical to support our conclusions. We think that showing that the protein is there at all is sufficient for now.

4. Fig. 6C, lines 238-239. Does the deletion of other cell division genes, (such as *ezrA*, which the authors used for comparative purposes), also result in mis-localization of GpsB, similar to the mis-localization of FtsZ shown in Fig. S3C? The authors are invoking a direct effect of the deletion of *facZ* in mis-localizing GpsB (which reportedly associates with FtsZ), but I wonder if a general defect in Z-ring assembly (similar to deletion of *ezrA*) also causes a mis-localization of GpsB and other divisome/FtsZ- associated proteins?

Response: Given that GpsB interacts with FtsZ, we imagine that any defect in Z-ring positioning would affect GpsB localization. The mislocalization of GpsB in the absence of FacZ is not the main result that we use to functionally connect the two proteins. Rather, the specific connection is strongly supported by the observation that *gpsB* mutations suppress FacZ defects and that the two proteins interact *in vitro*.

5. Fig. 6D-E. The authors invoke an interaction between GpsB and FacZ, but the subcellular localization of the two proteins are almost entirely non-overlapping: GpsB appears to localize at mid-cell with FtsZ, but FacZ is present on the cell periphery and the periseptal region. Can the authors speculate why two proteins that interact robustly in vitro would not largely co-localize in vivo?

Response: The reviewer raises a good point. We have added microscopy analysis (demographs) showing FacZ and GpsB localization pairwise within single cells. In early division, when the Z-ring is open, FacZ and GpsB localization is overlapping. Later, as the Z-ring constricts, GpsB closes with the ring, but FacZ is left behind (Fig 6A-B). This localization pattern may be related to the observation that FtsZ and FacZ utilize the same motif to interact with GpsB, with competition between FacZ and FtsZ for binding to GpsB preventing FacZ from completely mirroring GpsB localization.

6. Lines 249-251, lines 270-271. I am a bit confused by the model as presented. The authors state that FacZ ‘prevents aberrant envelope invaginations’, but also imply that FacZ only localizes once ‘membrane invagination is initiated’. Please clarify. On a more semantic note, although it is clear from the data that FacZ is an important regulator of Z- ring formation, is it proper to refer to a late-divisome protein (which localizes only after cell division initiates) as a ‘division site placement factor’ in the Title? The abstract also suggests that FacZ can control ‘division site placement in *S. aureus* and other Firmicutes’, but it appears that *B. subtilis* (Fig. S5B) finds mid-cell efficiently even in the absence of FacZ, arguing against a role in division site placement, per se. Consider softening the conclusion and the title with respect to ‘placement’, especially since other important cell division factors like EzrA also show a similar Z-ring assembly defect when removed and are not referred to as ‘placement factors’.

Response: The reviewer raises an interesting point. We argue that any factor required to prevent multiple, aberrantly placed invaginations or septa qualifies as a “division placement factor”, and we would support EzrA being referred to as a Z-ring placement factor even if it has not been traditionally called as such. The term “division placement factor” does not require or imply that the factor must arrive to a location before FtsZ, only that division sites are aberrantly placed in its absence. For example, the MinCD proteins in *B. subtilis* are late recruits to the septum, yet they have been implicated in preventing new division sites near completed septa. Should we not call them division placement factors since they are late recruits? We feel the term “division placement factor” is sufficiently broad to encompass FacZ and have therefore kept this designation. Prompted by the points raised by the reviewers, in the revised manuscript we make this point more explicitly. We indicate that FacZ functions differently than “traditional” placement factors. The Summary now reads:

"Thus, FacZ is a novel division site placement factor in *S. aureus* with an unconventional mode of action. Rather than acting at the level of FtsZ, it antagonizes the activity of GpsB, an FtsZ-

binding protein in *S. aureus* that forms an interaction hub at the division site connecting several envelope biogenesis factors with the cytokinetic ring.”

Minor comment

7. Line 273. The authors describe the junction of the septal and peripheral membranes as ‘positive’ Gaussian curvature, but since the principal normal curvatures on the cytoplasmic face of the membrane curve in opposite directions, I believe this should be ‘negative’ Gaussian curvature (as measured from inside the cell, where FacZ localizes), correct?

Response: An observer inside of the cell, standing with their feet on the cytoplasmic face of the periseptal membrane with their head in the cytoplasm, would find that the membranes curved in the same direction (up, or positive) whichever direction they looked. Since Gaussian curvature is the product of the two principle curvatures, its sign is positive. More importantly, Gaussian curvature is independent of the observer's position with regard to the surface in question. If an observer were standing on the opposite (exoplasmic) face of the membrane, both principle curvatures would curve "down" away from the observer (negative). Their product, however, would still be positive.

Reviewer #3 (Remarks to the Author):

This paper presents innovative variations of cell sorting of transposon-mutagenized cells of *Staphylococcus aureus* to catalog insertions in non-essential genes that affect cell size/clustering and/or membrane permeability. The screening methods are validated and compared. As a control, they turn up known genes in cell envelope synthesis and assembly as well as a number of new genes, which are listed in the figures and supplemental tables. The paper then focuses on one of these new genes, *facZ*, which has substantial effects on cell size, division/clumping, and membrane permeability. The paper goes on to characterize some phenotypes and genetic interactions of *facZ*. Δ -*facZ* mutants have unusual phenotypes of forming clumps of larger cells with membrane and peptidoglycan mislocalized into foci adjacent to nucleoids. These phenotypes are complemented in control experiments. *facZ* homologs are found in some, but not all Firmicutes, and Δ -*facZ* mutants of *B. subtilis* also have some defects that parallel those in *S. aureus*. The authors then go on to characterize additional genetic relationships of *facZ* in *S. aureus*. They show that localization of FtsZ is perturbed in a *facZ* mutant in a way reminiscent of in *ezrA* mutants. Fusion to mCherry shows that FacZ localizes around the cell membrane with concentration to the periseptal region of dividing *S. aureus* cells. Tn-seq and follow-up experiments show that *EzrA* and *FacZ* are synthetically lethal and that mutants lacking *EzrA* or *FacZ* are hypersensitive to the PL19 compound that inhibits FtsZ GTPase in many Gram-positive bacteria. Null mutations in *gpsB* suppressed the hypersensitivity of Δ -*facZ* mutants to PC19, and Δ -*gpsB* mutations restored a near-WT cell phenotype to Δ -*facZ* mutants. In addition, *FacZ* was required for normal septal localization of *GpsB*. Last, *FacZ* contains a known binding motif for *GpsB*, and a repeated version of this binding motif from *FacZ*

26bound to a domain of GpsB. Together, these results lead to model where FacZ concentrates to the periseptum and localizes and restricts GpsB activity to the septal region, thereby preventing aberrant GpsB activity around *S. aureus* cells.

This paper presents a useful screen and new candidates with roles in cell surface biogenesis in *S. aureus*. The screen is convincingly validated, and the new candidates will be the subject for new studies. In addition, the study of FacZ function presents many interesting new results that establish FacZ as an important new division protein in *S. aureus* through positioning of the GpsB regulator. The paper is well written and

convincing for the most part. Overall, the data is of high quality and clearly presented. However, there are a few topics that require greater context and explanation. Several simple experiments would strengthen some of the main conclusions, and the biochemical experiments are not complete enough. Some suggestions are included below that would improve this very interesting study.

Response: We thank the reviewer for their careful review of the manuscript and their helpful feedback. We appreciate the reviewer's comments on the innovative nature of the screen, the quality of the writing, and overall enthusiasm for the work.

1. Line 42-43, Fig. S4, and Discussion (line 288). There is no evidence yet for “a conserved role” in “other Firmicutes” (but there could be with simple additional experiments in point 2, below). There is a sizable slice of the Firmicute pie missing in Fig. S4. Please specify which Firmicute groups do not have FacZ, but still contain GpsB? This would provide context on how conserved this role is. Along this line, line 43 should be changed to “and certain other Firmicutes.” It would help to redraw Fig. S4 larger and with more detail to indicate the Firmicute groups with and without FacZ and whether any also lack GpsB. It is also curious that Actinobacteria also contain FacZ. Do they also contain a GpsB homolog? Please consider mentioning this topic very briefly in the Discussion.

2. To generalize the mechanism more, does delta-gpsB reverse the delta-facZ phenotypes in *B. subtilis* (Fig. S5). Along the same line, does FacZ in *B. subtilis* and the other selected Firmicutes in Fig. S4 contain a conserved GpsB binding site? Also, is a delta-facZ *B. subtilis* mutant hypersensitive to PC 19 and synthetically lethal with delta- *ezrA*. Results from these simple experiments would be highly informative that the mechanism in *S. aureus* extends to *B. subtilis*.

3. One other point of context that should be mentioned in the Discussion is that FtsZ from *S. aureus* is unusual in that it contains a special C-terminal tail that binds to the N-terminal domain of GpsB (Sacco et al. in references). This tail is not found in FtsZ homologs from some other species. Therefore, there would seem to be a competition for GpsB binding between FtsZ and FacZ (and PBP4) that does not occur in other species, including *B. subtilis*. This competition could play into the FacZ mechanism reported here and may be unique to Firmicute species like *S. aureus* that have this type FtsZ-GpsB interaction.

Response to points 1-3: These three points all address the potential conservation of FacZ function among the Firmicutes, and we appreciate the reviewer pressing us on

this issue.

Prompted by the points raised by the reviewer, we performed additional experiments related to FacZ function in *B. subtilis*. These experiments revealed that $\Delta facZ$ and $\Delta gpsB$ mutants are both PC19 sensitive in *B. subtilis*, and the double mutant does not rescue this sensitivity (Fig S7I). Furthermore, inactivation of GpsB does not suppress the membrane defects of a $\Delta facZ$ mutant in *B. subtilis*. These data argue that that

precise functions of *S. aureus* FacZ and *B. subtilis* FacZ are less similar than we originally proposed. As the reviewer rightly points out, *B. subtilis* FacZ and FtsZ both lack the GpsB-binding motifs that are present in *S. aureus*. Thus, although FacZ does appear to be important for proper Z-ring and septum formation in *B. subtilis* and the *B. subtilis* mutant has membrane defects that resemble those in the $\Delta facZ$ *S. aureus* mutant, it is likely the two proteins function via different molecular mechanisms. One possibility is that *B. subtilis* FacZ interacts with a different division protein. We now discuss the potential differences between FacZ function in *S. aureus* and *B. subtilis* in the revised manuscript and have softened statements on the conservation of function to indicate that although they both may function in Z-ring/septum placement, they are likely working via different underlying mechanisms. Accordingly, we have also changed the descriptor of the FacZ name to be “Factor preventing extra Z-rings”, which eliminates the implication it works the same throughout Firmicutes. We think that FacZ represents an interesting example of how division factors may be repurposed or perhaps more appropriately “multi-purposed” in pathogens with simplified morphologies and genomes relative to rod-shaped soil organisms.

We thank the reviewer for pushing us to do more experiments related to the *B. subtilis* protein.

Finally, we have added a figure panel to more clearly show the extent of FacZ conservation within the Firmicutes and changed the text as suggested.

4. Line 133 and line 50 in Supplemental. It's stated that the PCA correlated genes have well known functions. It would help readers outside of *S. aureus* to list gene functions, possibly in Fig. S1, where there is room for an addition.

Response: A list of functions has been added as suggested. See figure S1G.

5. Line 144. What are the growth properties of the delta-facZ mutant in *S. aureus*? Do the cells grow more slowly and to a lower final OD, and do the cells remain viable in exponential and stationary phase? Since this is the first report of this gene, this information is of interest. Also, the micrographs seem to indicate a considerable heterogeneity of cell sizes of the delta-facZ mutant that is not fully reflected in the graphs of cell areas. Please clarify this point briefly in the text.

Response: A $\Delta facZ$ mutant grows a bit slower than wild-type, most likely due to cell

lysis as seen in PI staining. This slow growth can be observed in spot titers where colonies of the mutant are slightly smaller than those formed by wild-type cells (see S6F, right panel). We have included growth curves in the revision to better compare the growth rates between wild-type and $\Delta facZ$ cells (see Figure S5C).

The cell size heterogeneity of the $\Delta facZ$ mutant is reflected by the elongated tail extending upwards in the violin plots of cell area (Figures 3B, 5F). To document the reproducibility of this heterogeneity, we have included a graph showing the increased coefficient of variance of cell size of $\Delta facZ$ mutants (Figure S7D, right panel).

6. Line 182. You might mention that Fig. S6 shows that the FacZ-mCherry fusion is functional, which is an important point.

Response: This point is now explicitly stated in the text.

7. Figure 2A. Why is there only one enriched Tn insertion in the C-terminus of *facZ*? Is there only one TA in *facZ* or could this mean that the *facZ* phenotypes only appear when the C-terminus is inactivated. Please clarify.

Response: The *facZ* gene has many TA sites. We think the result from the Tn-Seq experiment is related to the small size of the gene. In our experience small targets often have fewer unique insertions in the initial library. This was the case with *facZ*. The phenotype of the deletion mutant shows that the phenotype is due to the loss of FacZ, not just the production of a protein lacking the C-terminus. Notably, we have added experiments that show that inactivation of the C-terminal GpsB-binding motif is sufficient to disrupt FacZ function, so C-terminal transposons are likely to be inactive, consistent with the Tn-seq results.

8. Line 204. Table S6 seems to show that there are other genes that are synthetically lethal with FacZ. Is this correct, and if so, it should be commented on. Also, based on the previous publications about EzrA function in *S. aureus*, is FtsZ-ring formation the only reason underlying the synthetic lethal relationship between FacZ and EzrA. This part of the story seems to drop off. Does delta-*gpsB* also suppress the hypersensitivity of delta-*ezrA* to PC 19 or are delta-*gpsB* and delta-*ezrA* synthetically lethal in *S. aureus*? Please expand slightly.

Response: The reviewer is correct. There are many synthetic lethal partners for *facZ*. We now discuss this point in the revised text (see response to comment 1c of Reviewer 1). To expand a little more on the *ezrA-facZ* genetic connection, we analyzed viability by spot-titer in cells depleted of EzrA in WT, $\Delta facZ$, $\Delta gpsB$, or $\Delta facZ \Delta gpsB$ mutant backgrounds (see Figure S5I). Interestingly, $\Delta gpsB$ is not synthetically lethal with EzrA depletion. Notably, $\Delta gpsB$ partially rescues the synthetic lethality of EzrA depletion in cells lacking FacZ. These results suggest that the $\Delta ezrA \Delta facZ$ synthetic lethality is related to problems with Z-ring formation. These data are discussed in the revised manuscript.

9. Line 219. Were other suppressors of PC19 hypersensitivity of delta-*facZ* identified in genes other than *gpsB*? If so, which genes were they? The identity of other suppressors should be listed and could be highly relevant to the model.

32

Response: Yes, other suppressors of the PC-19 sensitivity of $\Delta facZ$ mutants were isolated. We have not included them because they require further characterization that is beyond the scope of this report. Thus far, the only suppressors in factors directly connected to FacZ are those in *gpsB*.

10. Line 228. It is remarkable that delta-gpsB seems to have minimal phenotypes (Fig.S7). This point might be emphasized more. The results of delta-gpsB alleviating the delta-facZ defects is very nice, as is the demonstration that GpsB localization is dependent on FacZ in *S. aureus*.

Response: We agree that the minimal phenotypes associated with the Δ *gpsB* mutant is a striking result. We emphasize this in the Results section by comparing it to contrasting results that previously suggested *gpsB* is essential. We also comment on the dispensability of the FacZ-GpsB module in the Discussion section.

11. Line 241 and Fig. 6D. The biochemical experiments are incomplete. An appropriate experiment was done showing that three tandem repeated copies of the putative GpsB binding site from FacZ fused to SUMO binds to the N-terminal domain of GpsB. However, Fig. 6D should be relabeled, because “FacZ” was not really used. Does a single copy of the binding site, as in the native protein (Fig. S6), fused to SUMO also bind to the GpsB domain? Most definitively, does the purified cytoplasmic domain of FacZ bind to GpsB? These experiments should be tried and included, if they can be done, to really indicate direct binding between FacZ and GpsB. The GpsB binding sites might be indicated on the predicted models of FacZ in Fig. 6B and 6C. From the numbering in the text, it seems like the GpsB binding motif is in the prongs of the “anchor” structure.

Response: We have added additional support for the GpsB-FacZ interaction as requested. First, tagged versions of GpsB and FacZ were co-expressed in *S. aureus*, and co-immunoprecipitation experiments indicate that the two proteins reside in a complex *in vivo* (Fig S6E). We also tested whether a single iteration of FacZ’s GpsB-binding motif was sufficient for an interaction with GpsB. It was not. Given that GpsB is a proposed hexamer and our AlphaFold modeling predicts that FacZ also forms homo-oligomers, we suspect that *in vivo* binding is likely driven by avidity rather than affinity. Attempts were made to purify the cytoplasmic portion of FacZ to test whether oligomers of FacZ can interact with GpsB hexamers; however, cytoplasmic FacZ formed insoluble aggregates, preventing us from performing these interaction studies.

To further characterize the interaction, arginine residues in the predicted GpsB-binding region of FacZ were replaced with aspartic acids, and we present evidence that the altered FacZ protein is non-functional in cells (Fig. S6C-E) and that the tandem binding motifs with these changes fail to interact with GpsB in our SEC assay (Fig. 6A, S6F). We have updated the figures with these new data, and now address it in the Discussion. We have also updated the figure labels as the reviewer suggested. We thank the

reviewer for requesting additional experiments. We think the new data strengthens our conclusions.

12. Line 264. Please qualify to: “interact directly with FtsZ in *S. aureus* and to promote...”, because GpsB binding to FtsZ has not been detected in other Gram-positive species (please see point 3, above).Response: Corrected as suggested.

Reviewer #4 (Remarks to the Author):

This paper does a nice job of finding and describing a novel nonessential protein that influences cell division. A molecular mechanism for how this occurs is unclear but it appears to somehow restrict Z ring number or orientation.

This paper uses flow cytometry to find genes in *S aureus* involved in cell morphology/envelope integrity. Using this approach, they find expected genes as well as genes of unknown function. They pick one of these, designated FacZ, and examine it in some detail because of the curious envelope defects/morphology that includes internal membrane deposition that is accompanied with cell wall deposition and connected to the peripheral envelope. A knockout of FacZ leads to altered morphology which they trace back to spurious/multiple Z rings. Interestingly, a knockout of GpsB, which has little phenotype, appears to almost completely reverse the FacZ phenotype. Thus, deletion of two nonessential genes cancel each other out leading one to wonder what they are doing.

The first part of the paper describes and validates the two screens that were used – mutants with increased light scattering and those with increased uptake of a dead cell dye (Figs.1 & 2 and S1 and S2 and Tables S1-6). The results indicate that the screens yield somewhat overlapping genes. Since, they are screening for underrepresented genes from random transposon libraries that give the above phenotypes, they are looking for nonessential genes that affect morphology/envelope integrity. The description of the screen and the reporting of the results are very thorough.

Response: We thank the reviewer for their enthusiastic response to our study and for providing helpful feedback.

1) Further study of FacZ show it's a peripheral membrane protein not enriched at the septum. However, FacZ is connected to division since it is synthetic lethal with EzrA and like a EzrA mutant is sensitive to an FtsZ inhibitor. Curiously, the authors find that this sensitivity is alleviated by inactivating GpsB. Whereas GpsB appears to be mostly at the septum in WT cells, in a FacZ mutant it is associated with the unusual membrane depositions that led the authors to study FacZ in the first place. If GpsB is considered a septal protein then this argues that these

depositions are septal material. This should be confirmed by examining the localization of an essential protein to verify that it is septal synthesis – any of FtsQ, L, B, W or I would suffice.

Response: To determine whether the envelope invaginations are caused by misplaced divisome factors and inappropriate septation, we monitored the localization of GFP fusions to PBP1 (FtsI) and FtsW, as the reviewer recommended. We found that both proteins are mislocalized in a $\Delta facZ$ mutant, and that they are located at or near the aberrant envelope invaginations. These results are consistent with the model that these

invaginations are misplaced septal material and thus represent aberrant division attempts (see Fig. S3 E-F). We thank the reviewer for suggesting this experiment; we think it strengthens our conclusions.

Other comments:

2) Fig. 3A. How come the FacZ mutant lacks the red halo from the membrane stain?

Response: Mutants inactivated for FacZ have peripheral membrane, which is essential for survival. However, the envelope foci are larger and brighter than the peripheral cell envelope. The image brightness and contrast have been adjusted to promote easy visualization of the membrane defects, which makes visualizing the peripheral membranes difficult. This adjustment was especially necessary when the color channels were merged, making it difficult to see subtle signals. Elsewhere in the paper (Figure 5E, S3E-F, S5A) we show images of $\Delta facZ$ mutant membranes in greyscale. They are scaled so that the defects and peripheral membranes are easily visualized at the same time.

3) Fig. S4. I assume the blue indicates species that have the gene. This means more species lack the gene than have it?

Response: The tree (now moved to Fig. S7A-B) shows the conservation of FacZ homologs in all bacteria, not just Firmicutes. FacZ is not conserved in all Firmicutes, and is sometimes found elsewhere, notably the Actinobacteria. The conservation of FacZ within the Firmicutes, which is 74.6%, or 2042 of the 2737 Firmicute genomes used to construct this phylogeny in AnnoTree. In Actinobacteriota, the second-most FacZ-enriched monophyletic clade (with over 100 genomes), 28.4% (1211/4261) have a FacZ homolog.

4) Lines 190-202. Description of FacZ localization. In line 201 it is concluded that “FacZ is enriched at midcell”. However, on lines 193-4 it is stated that the intensity profile is “as expected for a membrane protein”. I think I understand what the authors are getting at but it is a bit confusing.

Response: The description of the localization was poorly communicated. The description of FacZ localization has been modified. Please see responses to Comment 2b of Reviewer 1 and Comment 2 of Reviewer 2.

5) Fig. S7D. What is the size of FacZ based on elution volume? Monomer or multimer? Only full length FacZ elutes with GpsB – the break down products do not. Is it known if these are missing the N or C terminus?

Response: We apologize for the confusion. The purified FacZ protein used in the size exclusion chromatography is a tandem repeat of 3 copies of the C-terminus of FacZ (amino acid 127-146). We have modified the labels in the figures in the main text and supplement to make this more clear. They are now labeled as SUMO-3X(FacZ¹²⁷⁻¹⁴⁶).

The reviewer is correct, only the full-length SUMO-3X(FacZ¹²⁷⁻¹⁴⁶) eluted with GpsB. We have not determined whether the break down products are missing the N or C terminus, but we suspect these are missing part of the repeats from the C terminus.

The sizes of SUMO-3X(FacZ¹²⁷⁻¹⁴⁶), GpsB, and the complex based on SEC elution volume are imprecise, but our size standards suggest GpsB is likely to be a dimer and the elution profile of the GpsB/SUMO-3X(FacZ¹²⁷⁻¹⁴⁶) complex suggests that only one SUMO-3X(FacZ¹²⁷⁻¹⁴⁶) interacts with the GpsB dimer.

Decision Letter, first revision:

Message: Our ref: NMICROBIOL-23040978A

20th November 2023

Dear Tom,

Thank you for your patience as we've prepared the guidelines for final submission of your Nature Microbiology manuscript, "Identification of FacZ as a division site placement factor in *Staphylococcus aureus*" (NMICROBIOL-23040978A). Please carefully follow the step-by-step instructions provided in the attached file, and add a response in each row of the table to indicate the changes that you have made. Please also check and comment on any additional marked-up edits we have proposed within the text. Ensuring that each point is addressed will help to ensure that your revised manuscript can be swiftly handed over to our production team.

39We would like to start working on your revised paper, with all of the requested files and forms, as soon as possible (preferably within two weeks). Please get in contact with us if you anticipate delays.

In recognition of the time and expertise our reviewers provide to Nature Microbiology's editorial process, we would like to formally acknowledge their contribution to the external peer review of your manuscript entitled "Identification of FacZ as a division site placement factor in *Staphylococcus aureus*". For those reviewers who give their assent, we will be publishing their names alongside the published article.

Nature Microbiology offers a Transparent Peer Review option for new original research manuscripts submitted after December 1st, 2019. As part of this initiative, we encourage our authors to support increased transparency into the peer review process by agreeing to have the reviewer comments, author rebuttal letters, and editorial decision letters published as a Supplementary item. When you submit your final files please clearly state in your cover letter whether or not you would like to participate in this initiative. Please note that failure to state your preference will result in delays in accepting your manuscript for publication.

Cover suggestions

COVER ARTWORK: We welcome submissions of artwork for consideration for our cover. For more information, please see our [guide for cover artwork](https://www.nature.com/documents/Nature_covers_author_guide.pdf).

Nature Microbiology has now transitioned to a unified Rights Collection system which will allow our Author Services team to quickly and easily collect the rights and permissions required to publish your

work. Approximately 10 days after your paper is formally accepted, you will receive an email in providing you with a link to complete the grant of rights. If your paper is eligible for Open Access, our Author Services team will also be in touch regarding any additional information that may be required to arrange payment for your article.

Please note that *Nature Microbiology* is a Transformative Journal (TJ). Authors may publish their research with us through the traditional subscription access route or make their paper immediately open access through payment of an article-processing charge (APC). Authors will not be required to make a final decision about access to their article until it has been accepted. [Find out more about Transformative Journals](https://www.springernature.com/gp/open-research/transformative-journals)

Authors may need to take specific actions to achieve [compliance](https://www.springernature.com/gp/open-research/funding/policy-compliance-faqs) with funder and institutional open access mandates. If your research is supported by a funder that requires immediate open access (e.g. according to [Plan S principles](https://www.springernature.com/gp/open-research/plan-s-compliance)) then you should select the gold OA route, and we will direct you to the compliant route where possible. For authors selecting the subscription publication route, the journal's standard licensing terms will need to be accepted, including [self-archiving policies](https://www.nature.com/nature-portfolio/editorial-policies/self-archiving-and-license-to-publish). Those licensing terms will supersede any other terms that the author or any third party may assert apply to any version of the manuscript.

Reviewer #1:

Remarks to the Author:

All in all I find that the authors have responded satisfactorily to the points raised. My only remaining concern is relating to the proposed role of FacZ being a division site placement factor as perturbed Z-rings can be caused by mutations in genes that we do not normally refer to as such. Examples are mutations in the teichoic biosynthesis genes.

Minor things: ftsK listed twice in Fig. S1E; Line 230 delete 'which'.

Reviewer #2:

Remarks to the Author:

I have no further suggestions for the authors.

Reviewer #3:

Remarks to the Author:

The authors are to be commended for doing an outstanding job of effectively addressing all of the comments from the previous four reviews. The authors addressed numerous points of interpretation, provided additional context and clarification when needed, and performed several additional experiments to address previous issues or to strengthen conclusions. They considerably strengthened an already strong paper. The revised version is a major contribution to this field.

Reviewer #4:

Remarks to the Author:

The authors have gone to great lengths to respond to the critiques by amending the manuscript and including new experiments including an experiment looking at the localization of FtsW and PbpB1 supporting the idea that mislocalized FtsZ is able to cause septal PG synthesis. I am satisfied with the revision. The results show that FacZ is an antagonist of GpsB in Staph. Interestingly, GpsB only becomes an issue when FacZ is deleted (there is little phenotype in WT cells when GpsB is deleted). Consistent

4with what is found in other bacteria, genes involved in spatial regulation of FtsZ are generally nonessential. The authors still call the gene FacZ even though evidence shows it is an antagonist of GpsB, however, the name is consistent with the phenotype.

Final Decision Letter:

Mess 15th January 2024

age:

Dear Tom,

I am pleased to accept your Article "FacZ is a GpsB-interacting protein that prevents aberrant division-site placement in *Staphylococcus aureus*" for publication in Nature Microbiology. Thank you for having chosen to submit your work to us and many congratulations.

Acceptance of your manuscript is conditional on all authors' agreement with our publication

5policies (see <https://www.nature.com/nmicrobiol/editorial-policies>). In particular your manuscript must not be published elsewhere.

Please note that *Nature Microbiology* is a Transformative Journal (TJ). Authors may publish their research with us through the traditional subscription access route or make their paper immediately open access through payment of an article-processing charge (APC). Authors will not be required to make a final decision about access to their article until it has been accepted. [Find out more about Transformative Journals](https://www.springernature.com/gp/open-research/transformative-journals)

Authors may need to take specific actions to achieve [compliance](https://www.springernature.com/gp/open-research/funding/policy-compliance-faqs) with funder and institutional open access mandates. If your research is supported by a funder that requires immediate open access (e.g. according to [Plan S principles](https://www.springernature.com/gp/open-research/plan-s-compliance)) then you should select the gold OA route, and we will direct you to the compliant route where possible. For authors selecting the subscription publication route, the journal's standard licensing terms will need to be accepted, including [self-archiving policies](https://www.nature.com/nature-portfolio/editorial-policies/self-archiving-and-license-to-publish). Those licensing terms will supersede any other terms that the author or any third party may assert apply to any version of the manuscript.

To assist our authors in disseminating their research to the broader community, our SharedIt initiative provides you with a unique shareable link that will allow anyone (with or without a

subscription) to read the published article. Recipients of the link with a subscription will also be able to download and print the PDF.

Congratulations once again and I look forward to seeing the article published.

With kind regards,

P.S. Click on the following link if you would like to recommend Nature Microbiology to your librarian <http://www.nature.com/subscriptions/recommend.html#forms>

** Visit the Springer Nature Editorial and Publishing website at http://editorial-jobs.springernature.com?utm_source=ejp_NMicro_email&utm_medium=ejp_NMicro_email&utm_campaign=ejp_NMicro for more information about our career opportunities. If you have any questions please click [here](mailto:editorial.publishing.jobs@springernature.com).**